# SAME: Uncovering GNN Black Box with Structure-aware Shapley-based Multipiece Explanation

**Ziyuan Ye[1,2,*], Rihan Huang[1,3,*], Qilin Wu[1,4], Quanying Liu[1,†]**
[1]Southern University of Science and Technology, [2]The Hong Kong Polytechnic University,
[3]King Abdullah University of Science and Technology, [4]Carnegie Mellon University
{ziyuanye9801, rihanhuang.work, kyrinwu}@gmail.com, liuqy@sustech.edu.cn

## Abstract

Post-hoc explanation techniques on graph neural networks (GNNs) provide economical solutions for opening the black-box graph models without model retraining. Many GNN explanation variants have achieved state-of-the-art explaining results on a diverse set of benchmarks, while they rarely provide theoretical analysis for their inherent properties and explanatory capability. In this work, we propose Structure-Aware Shapley-based Multipiece Explanation (SAME) method to address the structure-aware feature interactions challenges for GNNs explanation. Specifically, SAME leverages an expansion-based Monte Carlo tree search to explore the multi-grained structure-aware connected substructure. Afterward, the explanation results are encouraged to be informative of the graph properties by optimizing the combination of distinct single substructures. With the consideration of fair feature interactions in the process of investigating multiple connected important substructures, the explanation provided by SAME has the potential to be as explainable as the theoretically optimal explanation obtained by the Shapley value within polynomial time. Extensive experiments on real-world and synthetic benchmarks show that SAME improves the previous state-of-the-art fidelity performance by 12.9% on BBBP, 7.01% on MUTAG, 42.3% on Graph-SST2, 38.9% on Graph-SST5, 11.3% on BA-2Motifs and 18.2% on BA-Shapes under the same testing condition. Code is available at https://github.com/same2023neurips/same.

## 1 Introduction

Graph neural networks (GNNs) have demonstrated a powerful representation learning ability to deal with data in non-Euclidean space. However, the explanation techniques for deep learning models on images and text cannot directly apply to understand GNNs [27, 18, 6, 1]. There is a gap in the understanding of how GNNs work, which largely limits GNNs' application in many fields.

Many GNN explanation techniques aim to examine the extent to which GNNs depend on individual nodes and edges of the graph [22, 35, 19, 24, 37]. However, graph features within nodes and edges contribute different amounts of information when considered individually than contextualized with topology [37, 39]. Therefore, discovering the most important explanation with one or more connected components given an input graph and a well-trained GNN raises the additional challenge of handling structure-aware feature interactions. Recently, a number of studies, including GNN-LRP [25], SubgraphX [38] and GStarX [40] have endeavored to address this issue to some extent. Although many of the current GNN explanation techniques have empirically achieved state-of-the-art

---

[*]Equal contribution, co-first author.
[†]Corresponding author.

37th Conference on Neural Information Processing Systems (NeurIPS 2023).

explainability performance, the design of new GNN explanation techniques mainly relies upon empirical intuition, iterative experiments, and heuristics principles. To the best of our knowledge, the explanatory capability, potential limitations and inherent properties of GNN explanation techniques have not yet been thoroughly studied from a theoretical perspective.

In this work, we propose a novel Structure-Aware Shapley-based Multipiece Explanation (SAME) technique for fairly considering the multi-level structure-aware feature interactions over the graph by introducing expansion-based Monte Carlo tree search (MCTS). Our construction is inspired by the recently proposed perturbation-based GNN explanation methods [38, 40], which have proven effective for providing explainability from a cooperative game perspective. We summarize the main differences between SAME and previous work in Table 1. **The main contributions and novelties** of this work include the following. (1) *Theoretical aspect:* i) We review the characteristics of previous methods [22, 35, 19, 25, 38, 40] and highlight several desired properties (see Table 1) that can be considered by explanation methods for GNNs. ii) We provide the loss bound of the MCTS-based explanation techniques and further verify the superiority of our expansion-based MCTS in SAME compared to previous work [38] in an intuitive manner (Sec. 3.2). (2) *Empirical aspect:* Our experiments cover both real-world and synthetic datasets. The results show that i) SAME outperforms previous SOTA with fidelity and harmonic fidelity metrics under the same testing condition (Sec. 5.1). ii) SAME qualitatively achieves a more human-intuitive explanation compared to previous methods across multiple datasets (Sec. 5.2).

Table 1: Comparison of the properties of different GNN explanation techniques.

| Methods / Properties | Grad-CAM [22] | GNNExplainer [35] | PGExplainer [19] | GNN-LRP [25] | SubgraphX [38] | GStarX [40] | SAME(Ours) |
|---|---|---|---|---|---|---|---|
| Graph-level tasks | ✓ | ✓ | ✓ | ✓ | ✓ | ✓ | ✓ |
| Node-level tasks | | ✓ | ✓ | ✓ | ✓ | ✓ | ✓ |
| Feature interactions | | | | | ✓ | ✓ | ✓ |
| Structure awareness | | | | ✓ | ✓ | ✓ | ✓ |
| Multipiece explanation | ✓ | ✓ | ✓ | ✓ | | ✓ | ✓ |
| Node-wise importance | ✓ | | | | | ✓ | ✓ |
| Substructure-wise importance | | | | ✓ | ✓ | | ✓ |
| Composite-wise importance | | ✓ | ✓ | ✓ | ✓ | | ✓ |
| Priority-based integration | | ✓ | ✓ | | ✓ | | ✓ |
| Redundancy consideration | | ✓ | ✓ | | | ✓ | ✓ |

Note: *Feature interactions* and *structure awareness* are discussed in Sec. 3.1. *Multipiece explanation* is provided in Sec. 3.2. *Multi-grained importance (node / substructure / composite), priority-based integration* and *redundancy consideration* are presented in Sec. 4. Detailed mathematical definitions are provided in Appendix B.1.

## 2 Related Work

Improving the explainability of GNN models in a post-hoc fashion has been a theme in deep graph learning. An intuitive way to explain a well-trained GNN is to trace the gradient in the models [3, 22], where the larger gradient indicates the higher importance of node or edge of the graph. Previous work has also studied the decomposition-based methods [26, 25, 9] which decompose the final prediction into several terms and mark them as the important scores for input features. Another line of GNN explanation techniques lies in perturbation-based methods [19, 31, 11, 34] which usually obtain a mask for the input graph in various ways to identify the important input feature. SubgraphX [38], as one of a perturbation-based method, samples the subgraphs from the input graph by the pruning-based MCTS and finds the most important one via the Shapley value. However, the pruning-based MCTS in SubgraphX leads to a much larger search space and thus causes higher computational costs. Moreover, SubgraphX can only provide a single connected explanation for each graph, which limits its explanatory power in many scenarios that require multipiece explanation. Most recently, GStarX [40] scores node importance based on the Hamiache-Navarro (HN) value. Although GStarX also fairly considers the structure-aware feature interactions, it fails to account for the multi-grained importance, which might result in a suboptimal explanation. For other categories in GNN explanation techniques, including surrogate methods [30, 41, 8, 12], generation-based methods [36, 15, 32], and counterfactual-based methods [17, 2, 16], we refer readers to a recent survey [37].

# 3 Theoretical Motivation

**Notation and Preliminaries**. The well-trained target GNN to be explained can be formulated as $f : \mathcal{G} \to \mathcal{Y}$, where $\mathcal{G}$ denotes the space of input graphs and $\mathcal{Y}$ refers to the related label space. A graph can be denoted as $G = (V, X, E)$, where $V \in \mathbb{R}^{n \times n}$ represents node set, $X \in \mathbb{R}^{n \times d}$ is the node feature set and $E \in \mathbb{R}^{n \times n}$ denotes edge set. Given a well-trained GNN model $f(\cdot)$ and an input graph $G$, the goal of GNN explanation is to find the most important explanation $G_{ex}^*$ from $G$. Formally, this can be defined as an optimization problem that maximizes the importance of the explanation $G_{ex}$ for a given graph $G$ with $n$ nodes using an importance scoring function $I(f(\cdot), G_{ex}, G)$:

$$G_{ex}^* = \underset{G_{ex} \subseteq G}{\arg \max} \ I(f(\cdot), G_{ex}, G), \tag{1}$$

where each explanation $G_{ex}^i$ has $n_i$ nodes, and the other nodes not in the explanation can be expressed as $\{G \backslash G_{ex}^i\} = \{v_j\}_{j=n_i+1}^n$. It is noteworthy that each explanation $G_{ex}^i$ might contain one or more substructures (*i.e.*, connected components).

## 3.1 Structure-aware Shapley-based Explanations Satisfies Fairness Axioms

Unlike grid-like images or sequence-like texts, graphs have more complex and abstract structures. The importance scoring function of explanations determines the reliability and explanatory power of the GNN explanation method. Therefore, we present the idea that a comprehensive assessment of the importance of an explanation should consider the feature interactions under the constraints of the input graph's topology.

Shapley value [28], originating from cooperative game theory, is the unique credit allocation scheme that satisfies the fairness axioms. This concept is similar to the importance scoring function for explanation with the consideration of feature interactions. Some previous work have brought Shapley value into deep learning explanation methods [18, 5, 14]. In our study, the importance assessment of explanation is treated as a cooperative game, where the explanation $G_{ex}^i$ and all nodes not in the explanation $\{G \backslash G_{ex}^i\}$ are the players in the game. Therefore, when scoring the importance of any explanation $G_{ex}^i$, a set of players participating in the game can be denoted as:

$$P_i = \{G_{ex}^i, \underbrace{v_{n_i+1}, v_{n_i+2}, \dots, v_n}_{\{G \backslash G_{ex}^i\}}\}.$$

Inspired by the close connection between *feature interactions* and Shapley value, we define several desirable properties of importance scoring function for explanation according to fairness axioms:

**Property 1.** *(Efficiency). The sum importance of all players $p_j$ in $P_i$ is the same as the improvement of GNN $f(\cdot)$ on $P_i$ over an empty set, $\sum_{j=1}^{|P_i|} I(f(\cdot), p_j, G) = f(P_i) - f(\emptyset)$.*

**Property 2.** *(Symmetry). For any explanation $G_{ex}^i$, if there exist two other players $p_j, p_k \in \{P_i / G_{ex}^i\}$ that satisfy $f(G_{ex}^i \cup p_j) = f(G_{ex}^i \cup p_j)$, then $I(f(\cdot), p_j, G) = I(f(\cdot), p_k, G)$.*

**Property 3.** *(Dummy). If a player $p_j$ makes no contribution to GNN $f(\cdot)$, i.e. $f(G_{ex}^i \cup p_j) = f(G_{ex}^i)$ holds for any $G_{ex}^i$, then $I(f(\cdot), p_j, G) = 0$.*

**Property 4.** *(Monotonicity). Consider two well-trained GNN models $f_1(\cdot)$ and $f_2(\cdot)$, given an explanation $G_{ex}^i$, if for any player $p_j$, $f_1(G_{ex}^i \cup p_j) - f_1(p_j) \geq f_2(G_{ex}^i \cup p_j) - f_2(p_j)$ always holds, then $I(f_1(\cdot), G_{ex}^i, G) \geq I(f_2(\cdot), G_{ex}^i, G)$.*

Shapley value can well satisfy the above four fairness Properties 1-4. However, the Shapley-based importance scoring function attempts to take all nodes in the graph except explanation $G_{ex}^i$ into the cooperation game, which not only ignores the topological information of the input graph but also brings huge computational costs. Fortunately, [38] alleviate this issue by modifying the Shapley-based importance scoring function as 'k-hop Shapley', which also makes it *structure-aware*:

$$I(f(\cdot), G_{ex}^i, G) = \sum_{p_i \subseteq \{P_{i,khop} \backslash G_{ex}^i\}} \frac{|p_i|!(|P_{i,khop}| - |p_i| - 1)!}{|P_{i,khop}|!} (f(p_i \cup G_{ex}^i) - f(p_i)), \tag{2}$$

where $I(f(\cdot), G_{ex}^i, \mathcal{G})$ denotes the weighted sum of the marginal contribution of explanation $G_{ex}^i$, and $P_{i,khop} = \{G_{ex}^i, v_{n_i+1}, v_{n_i+2}, \ldots, v_{n_i+k_i}\}$ includes the nodes within the $G_{ex}^i$ as well as the nodes in the k-hop neighbors of $G_{ex}^i$.

## 3.2 Structure-aware Shapley-based Multipiece Explanation Provide Strong Explainability

The characteristics and properties within a graph or node tend to be *jointly* influenced by more than one high-order connected community of the graph. This implies that the appropriate explanation within this context requires the GNN explanation method to provide the multiple connected substructures simultaneously.

To solve the above challenge, we first define the mathematical formalization of the search processes on graphs by utilizing the MCTS-based GNN explanation methods. Then we propose a mathematically coherent framework to explore the explanatory power of MCTS-based GNN explanation methods using computational methods for the multi-armed bandit problem in the MCTS algorithm [20, 13].

Our mathematical framework is based on the hierarchical partitioning of the MCTS search space $\mathcal{X}$. More precisely, the smoothness of the MCTS search space $\mathcal{X}$ can be defined by the inequality $|f(x_i) - f(x_j)| \leq l(x_i, x_j)$, where $l(x_i, x_j)$ refers to the Lipschitz continuity between any two substructures $x_i$ and $x_j$ in $\mathcal{X}$. It serves as a critical metric to ascertain the boundedness of the change in the explanation method $f(\cdot)$ concerning the change in input substructures. Assuming that the function $f(\cdot)$ is Lipschitz continuous and the $l$ is given, an evaluation of the function $f(\cdot)$ at any point $x_t$ enables us to define an upper bounding function $B_t(x)$ for $f(\cdot)$. This upper bounding function can be refined after each evaluation of $f(\cdot)$:

$$\forall x \in \mathcal{X}, f(x) \leq B_t(x) \overset{def}{=} \min_{1 \leq s \leq t} [f(x_s) + l(x, x_s)], \tag{3}$$

where metric $l$ satisfies the following Assumptions 1-3. In the context of computational uncertainties associated with MCTS, the evaluation strategy in (3) offers the potential to describe specific numerical estimates within an undefinable space.

> **Assumption 1.** *(Local smoothness). There exists at least one stage-optimal substructure $x^\star \in \mathcal{X}$ of $f(\cdot)$ (i.e. $f(x^\star) = \sup_{x \in \mathcal{X}} f(x)$) and $\forall x \in \mathcal{X}, f(x^\star) - f(x) \leq l(x, x^\star)$ holds.*

> **Assumption 2.** *(Decreasing diameters). There exists a decreasing sequence $\delta(h) > 0$, such that for any depth $h \geq 0$ and for any cell $\mathcal{X}_{h,i}$ of depth $h$, $\sup_{\mathcal{X}_{h,i}} l(\mathcal{X}, \mathcal{X}_{h,i}) \leq \delta(h)$ holds.*

> **Assumption 3.** *(Well-shaped cells). There exists $\nu > 0$ such that for any depth $h \geq 0$, any cell $\mathcal{X}_{h,i}$ contains a l-ball of radius $\nu\delta(h)$ centered in $x_{h,i}$.*

With the given assumptions, the search space $\mathcal{X}$ can be partitioned into $K^H$ subsets (*i.e.*, cells) $\mathcal{X}_{h,i}$ using a $K$-ary tree of depth $H$, where $0 \leq h \leq H, 0 \leq i \leq K^{h-1}$. Based on this partitioning method, the MCTS process can be treated as the expansion of this $K$-ary tree. The root of $K$-ary tree (*i.e.*, cell $\mathcal{X}_{0,0}$) corresponds to the whole search space $\mathcal{X}$. Each cell $\mathcal{X}_{h,i}$ corresponds to a node $(h, i)$ of the tree, where $h$ denotes the depth of the tree and $i$ refers to the index. Each node $(h, i)$ possesses $K$ children nodes $\{(h + 1, i_k)\}_{1 \leq k \leq K}$ s.t. the associated cells $\{\mathcal{X}_{h+1,i_k}, 1 \leq k \leq K\}$ form a partition of the parent's cell $\mathcal{X}_{h,i}$. Consequently, expanding one node requires adding one of its $K$ children to the current tree, which corresponds to subdividing the cell $\mathcal{X}_{h,j}$ into $K$ children cells $\{\mathcal{X}_{h+1,j_1}, \ldots, \mathcal{X}_{h+1,j_K}\}$. Assume that there exist a decreasing sequence $\delta(h) \geq 0$ that satisfies $\sup_{\mathcal{X}_{h,i}} l(\mathcal{X}, \mathcal{X}_{h,i}) \leq \delta(h)$ for any $\mathcal{X}_{h,i}$. The decreasing sequence $\delta(h)$ ensures that each cell size reduces with increasing depth.

The search space $\mathcal{X}$ can be divided through the above partitioning method according to $\delta(h)$, with respect to $l$-open balls. Let $\mathbf{T}_t$ denote nodes of the current tree, and $\mathbf{L}_t$ denotes the incoming leaves of $\mathbf{T}_t$ to be expanded at round $t$. Recalling our earlier definition of $B_t(h)$ which was derived from the Lipschitz continuity, we can now generalize it to a new representation that connects to $\delta(h)$. Formally, it is expressed as:

$$b_{h,i} = f(x_{h,i}) + \delta(h). \tag{4}$$

Based on this, we now consider which nodes will be expanded during the search. Note that Assumption 2 implies that the b-value of any cell contains $x^\star$ upper bounds $f^\star$. In other words, for any cell $\mathcal{X}_{h,i}$ such that $x^\star \in \mathcal{X}_{h,i}$,

$$b_{h,i} = f(x_{h,i}) + \delta(h) \geq f(x_{h,i}) + l(x_{h,i}, x^\star) \geq f^\star. \qquad (5)$$

This means that a leaf $(h, i)$ of a $K$-ary tree will never be expanded if $f(x_{h,i}) + \delta(h) < f^\star$. Therefore, under this partitioning strategy, the only set of nodes that will be expanded could be defined as $I \overset{def}{=} \cup_{h \geq 0} I_h$, which could be stated as

$$I_h \overset{def}{=} \{nodes\{h, i\} \text{ such that } f(x_{h,i}) + \delta(h) \geq f^\star\}. \qquad (6)$$

In order to derive a loss bound, we now define a measure of the quantity of near-optimal states, called *near-optimality dimension*. For any $\epsilon > 0$, the set of $\epsilon$-optimal states can be defined as

$$\mathcal{X}_\epsilon := \{x \in \mathcal{X}, f(x) \geq f^\star - \epsilon\}. \qquad (7)$$

**Definition 1.** *($\eta$-near-optimality dimension). The $\eta$-near-optimality dimension is the smallest $d \geq 0$ such that there exists $C \geq 0$, for all $\epsilon > 0$, the maximum number of disjoint $l$-balls of radius $\eta\epsilon$ with centers in $\mathcal{X}_\epsilon$ is less than $C\epsilon^{-d}$.*

Definition 1 represents the number of near-optimal states for a function $f(\cdot)$ around its optimal solution. It is important to note that $d$ is not an intrinsic property of $f(\cdot)$ as we are packing near-optimal states using $l$-balls. Instead, it characterizes both $f(\cdot)$ and $l$ and depends on the constant. In order to relate this measure to the algorithmic details, we also need to correlate it with the characteristics of the partitioning, specifically the shape of the cells. That is, the near-optimality dimension $d$ is dependent on a particular constant, which will be determined in accordance with the parameter $\nu$ as defined in Assumption 3.

**Lemma 1.** *Let $d$ be the $\nu$-near-optimality dimension, and $C$ the corresponding constant. Then $|I_h| \leq C\delta(h)^{-d}$.*

Building upon Lemma 1, we further analyze the loss of the MCTS-based GNN explanation methods across $n$ iterations in Theorem 1.

**Theorem 1.** *Let us write $h(n)$ the smallest integer $h$ such that $C\Sigma_{l=0}^h \delta(l)^{(-d)} \geq n$, then the loss of the MCTS-based GNN explanation methods is bounded as:*

$$r_n \leq \delta(h(n)) \qquad (8)$$

The loss $r_n$ reflects the gap between the obtained and optimal explanations over $n$ iterations. We aim to bound this loss by $\delta(h(n))$, illustrating that refining the partition of the search space $\mathcal{X}$ reduces the loss, thus better approximating the optimal explanation. We provide a complete description of the mathematical properties and theorem proofs of the framework in Appendix B.2.

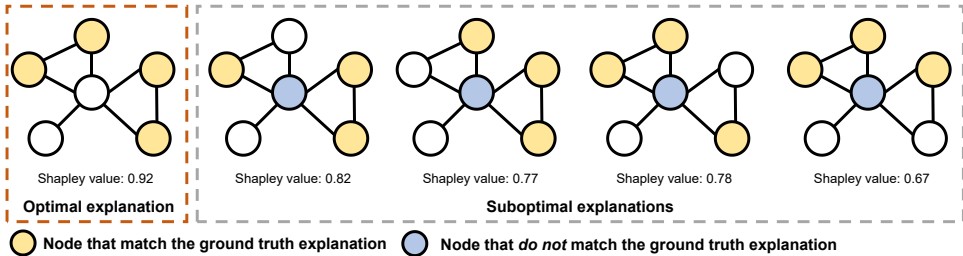

Figure 1: Illustration of ground truth explanation and the possible sub-optimal explanations provided by pruning-based MCTS explanation techniques.

Leveraging the mathematical underpinnings provided above, as demonstrated in Figure 1, we employ an example to contextualize the theoretical insights within the GNN explanation. The ground truth

explanation is highlighted in yellow which includes two connected components. When searching for explanations starting from any node in the two components through the pruning-based MCTS, *the nodes in other components* are accessible only via the unimportant node which is highlighted in blue. In this situation, given sparsity constraints, the pruning-based MCTS can only generate the suboptimal explanation. As a consequence, regardless of the search trajectory adopted, the diameter of the $l$-ball remains unyielding, failing to converge to the $\eta$-near optimality numerical solution. Therefore, it is necessary to design an explanation method that can accurately retain important nodes while avoiding irrelevant nodes, thus increasing the likelihood of discovering the optimal explanation. This ambition resonates with previously proposed loss bound $r_n \leq \delta(h(n))$, emphasizing the need for advanced exploration to reduce losses and approximate the optimal explanation with higher precision.

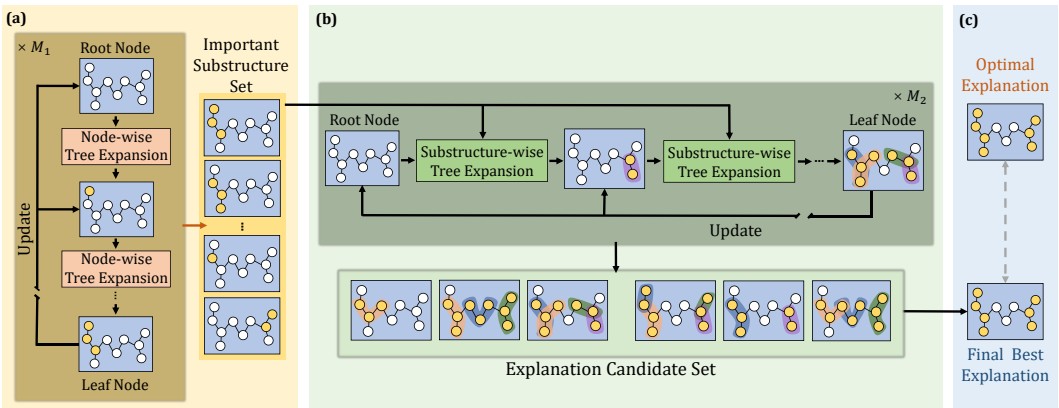

Figure 2: Overview of **S**tructure-**A**ware Shapley-based **M**ultipiece **E**xplanation (SAME) method. (a) *Important substructure initialization phase* aims at searching the single connected important substructure. (b) *Explanation exploration phase* provides a candidate set of explanations by optimizing the combination of different important substructures. (c) The comparison of the final explanation with the highest importance score from the candidate set with the optimal explanation.

# 4 Structure-aware Shapley-based Multipiece Explanation Method

As we discussed in Section 3, the structure-aware Shapley-based multipiece explanation provides a potential way to effectively uncover the GNN black box. In order to approximate the optimal Shapley-based explanation, we propose a two-phase framework, Structure-aware Shapley-based Multipiece Explanation (SAME) method, which is composed of (1) an *important substructure initialization* phase and (2) an *explanation exploration* phase, as shown in Figure. 2.

In the first phase (Section 4.1), we extend an expansion-based Monte Carlo tree search as an important substructure initializer to generate connected components not only of high importance but also of multi-grained diversity. In the second phase (Section 4.2), we apply the important substructure set as an action set in another expansion-based Monte Carlo tree search to explore potential explanations.

## 4.1 Important Substructure Initialization

In this section, we propose an expansion-based MCTS approach for important substructure initialization. It is intuitive that a favorable initialization should not exclude important substructures at any scale and should not include redundant substructures. Formally, these can be defined as:

**Property 5.** *(Node-wise importance). Given an input graph $G$ to be explained, for any node $v_i \in G$, its $I(f(\cdot), v_i, G)$ importance will be considered.*

**Property 6.** *(Substructure-wise importance). Given an input graph $G$ to be explained, for any substructure $G_{sub_i} \subseteq G$, its importance $I(f(\cdot), G_{sub_i}, G)$ will be considered.*

**Property 7.** *(Composite-wise importance). Given an explanation $G^i_{com} \subseteq G$ consisting of one or more substructures, its importance $I(f(\cdot), G^i_{com}, G)$ will be considered.*

**Property 8.** *(Priority-based integration). Given an explanation $G^j_{ex}$ with any size, the node $v_i \in \{G \backslash G^j_{ex}\}$ will be added on $G^j_{ex}$ to get a new explanation $G^k_{ex}$, if and only if for any $v_l \in \{G \backslash (G^j_{ex} \cup v_i)\}$, $I(f(\cdot), G^j_{ex} \cup v_i, G) > I(f(\cdot), G^j_{ex} \cup v_l, G)$ holds.*

**Property 9.** *(Redundancy consideration). Given an explanation $G_{ex} \subseteq G$, if $I(f(\cdot), G_{ex} \backslash \{i\}, G) > I(f(\cdot), G_{ex}, G)$ holds, the new explanation $G'_{ex} = G_{ex} \backslash \{i\}$ will be chosen.*

Taking the above properties into account, we propose our expansion-based Monte Carlo tree search for important substructure initialization which aims at providing important connected components of the graph. The detailed Algorithm is presented in Appendix C. Given a graph $G$ to be explained, the node in the MCTS is defined as $N_i$ which contains the following variables:

$$N_i : \{G^i_{sub}, T_i, R_i, A_i, C_i, W_i\}$$

where $G^i_{sub}$ denotes the corresponding substructure of graph $G$ for node $N_i$ in the search tree. $T_i$ is the visiting time of node $N_i$ in the search tree, $R_i$ refers to the the importance (reward) of substructure $G^i_{sub}$. $A_i$ represents the action set of node $N_i$. $C_i = \{N_j\}^{|A_i|}_{j=1}$ represents a set of children with respect to node $N_i$, and we denote $N_j$ as one of the child nodes obtained through the action $a_j \in A_i$ at parent node $N_i$. $W_i$ means the sum of the children's rewards.

The expansion-based MCTS is initialized with an empty set, $N_0 = \emptyset$. At the beginning of each iteration, our method will randomly choose an unvisited node from the graph, or choose the node with the highest reward if all nodes have been visited. Then, the tree will be iteratively expanded according to the ***node-wise tree expansion*** until it reaches a leaf node. Specifically, given an MCTS node $N_i$, it will only choose the child node within 1-hop neighbors of the associate $G^i_{sub}$ to expand, which means that the action set is topology dependent. Therefore, the substructure of any nodes in the search tree will be a connected component. During the child node selection, the reward of each child node $R_j$ of $N_i$ will be calculated following Equation (2). Finally, the chosen action is decided following the child selection strategy:

$$a^\star = \arg\max_{a_j} \ \frac{W_j}{T_j} + \beta R_j \frac{\sqrt{\sum_{k \in C_i} T_k}}{1 + T_j} \tag{9}$$

where $\beta$ is a hyperparameter for balancing exploration-exploitation trade-off [4]. We define the maximum substructure size as $\gamma$, and for any $G^i_{sub} \subseteq G$, $|G^i_{sub}| \leq \gamma$ always holds. After reaching the maximum substructure size, the reward of associated leaf nodes is backpropagated up the tree along the search path, updating all the information stored in each node of the path. The important substructure set includes all nodes in MCTS after performing $M_1$ times iterations.

### 4.2 Explanation Exploration

In this section, we propose another expansion-based MCTS for exploring high-explainable explanations. The detailed algorithm is provided in Appendix C. Different from the previous expansion-based MCTS provided in Section 4.1, we propose a slight modification on it such that we change the action set from node level to substructure level. As Section 3.2 discussed, this can be useful not only to obtain more flexible explanation results on real-world cases but also to provide the higher potential to approximate the theoretically optimal Shapley-based explanation.

The action set of MCTS in this phase is built upon the substructure set derived from the important substructure initialization phase. Similar to the previous MCTS, at the beginning of each iteration, it will randomly select an unvisited substructure from the set, or choose the substructure with the highest reward to expand if all substructures have been visited. Afterward, the ***substructure-wise tree expansion*** also follows the Equation (2) and (9) to develop the tree and provide the explanation candidate. The action set corresponding to each node of MCTS in the phase is the whole substructure set. To further accelerate the exploration, we filter the unimportant substructures by only keeping the top K important substructure in the action set.

Notice that calculating all possible combinations of either substructures or nodes can definitely obtain the optimal Shapley-based explanation. Nevertheless, such node-wise or substructure-wise brute force methods lead to $O(2^{|V|})$ computational complexity, which is an NP-hard problem. We provide a feasible solution to approximate the optimal Shapley-based explanation in polynomial time $\mathcal{O}(M_1\gamma|V|^2 + M_1\gamma|V| \times |E| + M_2 K t_s \frac{|V|\times(1-sparsity)}{\gamma})$, where $M_1$ and $M_2$ denote maximum number of iterations of the first phase and second phase respectively, $\gamma$ is the maximum substructure size, $K$ refers to the size of the important substructure set and $t_s$ is the time budget of MCTS in the second phase. We leave the derivation of the time complexity in Appendix B.4.

Table 2: Comparison of our SAME and other baseline using fidelity.

| Dataset | Graph classification | | | | | Node classif. |
|---|---|---|---|---|---|---|
| | Molecular graph | | Semantic graph | | Synthetic graph | |
| Methods | BBBP | MUTAG | Graph-SST2 | Graph-SST5 | BA-2Motifs | BA-Shapes |
| Grad-CAM [22] | 0.226±0.036 | 0.261±0.018 | 0.257±0.056 | 0.229±0.042 | 0.472±0.010 | - |
| GNNExplainer [35] | 0.148±0.041 | 0.188±0.031 | 0.143±0.041 | 0.170±0.046 | 0.442±0.026 | 0.154±0.000 |
| PGExplainer [19] | 0.197±0.043 | 0.156±0.004 | 0.219±0.040 | 0.207±0.036 | 0.431±0.011 | 0.135±0.020 |
| GNN-LRP [25] | 0.111±0.040 | 0.253±0.030 | 0.103±0.042 | 0.131±0.057 | 0.146±0.010 | 0.155±0.000 |
| SubgraphX [38] | 0.433±0.073 | 0.379±0.030 | 0.262±0.027 | 0.283±0.042 | 0.493±0.003 | 0.181±0.005 |
| GStarX [40] | 0.117±0.043 | 0.656±0.096 | 0.183±0.050 | 0.186±0.050 | 0.476±0.014 | - |
| **SAME** | **0.489±0.034** | **0.702±0.125** | **0.373±0.042** | **0.393±0.022** | **0.549±0.004** | **0.214±0.000** |
| **Relative Improve** | **12.9%↑** | **7.01%↑** | **42.3%↑** | **38.9%↑** | **11.3%↑** | **18.2%↑** |

Note: The fidelity results are averaged across different sparsity from 0.5 to 0.8. The quantitative results are presented in the form of mean ± std. The previous SOTA results on different datasets are marked with an underline. *Relative Improve* denotes the relative improvement of our SAME method over the SOTA methods.

## 5 Experiments

Our objective of the experiments is to understand the following two questions. 1) Are the explanations provided by SAME more informative and faithful compared to other methods under the same test conditions? 2) Can SAME provide a more human-intuitive explanation than others? To this end, we perform extensive quantitative and qualitative analysis to evaluate the explanatory power of SAME, following previous literature [38, 40]. The SAME is compared with various competitive baselines and shows state-of-the-art (SOTA) results in all the cases.

**Dataset**. The experiments are conducted on six datasets with diverse categories, including molecular graphs (e.g., BBBP [33] and MUTAG [7]), sentiment graphs (e.g., Graph-SST2 and Graph-SST5 [29]) and synthetic Barabási-Albert graphs (e.g., BA-2Motifs [19] and BA-Shapes [35]). We conduct a node classification task for the BA-Shapes dataset, and graph classification tasks for the rest five datasets. More detailed descriptions of datasets are provided in Appendix D.

**Metrics**. In this work, we use several criteria [37] to evaluate our approach: (1) *Sparsity* quantifies how compact are the explanations, and further facilitates fair comparison by restricting the different explanations to the same size. (2) *Fidelity* determines how informative and faithful are the explanations by removing the selected nodes. (3) *Inv-Fidelity* measures the explanations from the same aspect as fidelity while it keeps the selected nodes. (4) *Harmonic fidelity* [40] normalizes fidelity by sparsity and takes a harmonic mean to make different explanations comparable with a single metric. We leave detailed mathematical definitions of the above metrics in Appendix E.

**Experimental setup**. In the important substructure initialization phase, we set the MCTS iteration number $M_1$ to 20. The exploration-exploitation trade-off $\beta$ is set to 5 for BBBP and 10 for other datasets. The substructure size $\gamma$ has different settings in different datasets. In the explanation exploration phase, we set the hyperparameter $K = 7$ for important substructure filtering, $M_2 = 10$ for the MCTS iteration number, and the other hyperparameters of MCTS remain the same as the previous phase. We follow [37, 40] to set other baselines hyperparameters. All methods are implemented in PyTorch [21] and PyG [10]. Our experiments are conducted on a single Nvidia V100 GPU with an Intel Xeon Gold 5218 CPU. We leave the detailed settings in Appendix F.

Table 3: Comparison of inference time (in seconds) on different datasets.

| Methods \ Dataset | BBBP | MUTAG | Graph-SST2 | Graph-SST5 | BA-2Motifs | BA-Shapes |
|---|---|---|---|---|---|---|
| Grad-CAM [22] | 0.16 | 0.23 | 0.39 | 0.44 | 0.14 | - |
| GNNExplainer [35] | 7.56 | 1.96 | 7.64 | 19.39 | 1.89 | 2.72 |
| PGExplainer [19] | 0.15 | 0.21 | 0.35 | 0.43 | 0.12 | 0.13 |
| GNN-LRP [25] | 2.37 | 1.97 | 5.84 | 5.47 | 3.30 | 51.77 |
| SubgraphX [38] | 26.72 | 151.75 | 36.48 | 71.32 | 85.50 | 162.80 |
| GStarX [40] | 84.54 | 25.24 | 30.64 | 54.49 | 77.99 | - |
| **SAME** | 7.86 | 5.67 | 6.06 | 8.83 | 8.19 | 14.08 |

Note: The PGExplainer needs training before inferring the explanation.

Sentence: "a carefully structured scream of consciousness that is tortured and unsettling -- but unquestionably alive."

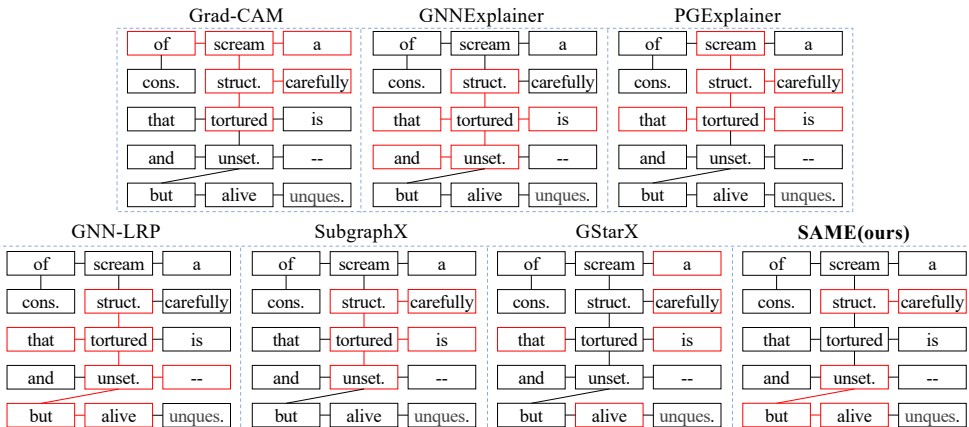

Figure 3: Comparison of the explanations on Graph-SST2 with GCN classifier.

## 5.1 Quantitative Analysis

To validate the overall explainability performance, we compare the proposed SAME with a series of competitive baselines under different metrics. Table 2 shows the averaged fidelity under different sparsity (*i.e.*, sparsity=[0.5,0.6,0.7,0.8]). The proposed SAME significantly outperforms the previous state-of-the-art on both real-world and synthetic datasets. Specifically, the performance improvement of SAME is 12.9% on BBBP, 7.01% on MUTAG, 42.3% on Graph-SST2, 38.9% on Graph-SST5, 11.3% on BA-2Motifs and 18.2% on BA-Shapes. Notably, we also demonstrate reliable improvements of SAME over previous SOTA methods in terms of harmonic fidelity at different sparsities, with an average improvement of 1.92% on the graph classification task. This result implies SAME has a higher ability to discover important components in the graph than all baseline methods. As for the inv-fidelity metric, the explanatory power of SAME is competitive compared with the previous SOTA. Detailed results for harmonic fidelity and inv-fidelity are in Appendix G.1. Since the proposed SAME is a model-agnostic method, it works well with GAT and GIN. More comparisons with GraphSVX [8] and OrphicX [16] are provided in Appendix G.3.

The computational cost of SAME and other baselines on different datasets are summarized in Table 3. We show that SAME consistently achieves much lower computational cost compared to GStarX and SubgraphX, reflecting its efficiency and robustness. This further verifies that expansion-based MCTS in SAME can work effectively in various scenarios.

## 5.2 Qualitative Analysis

Figure 3 presents the visualization comparison of the explanations on sentiment graphs. The nodes of adjectives and adverbs are considered to be important for they reveal the attitude of a sentence. Thus, the optimal explanation here is therefore the word or phrase with a positive meaning. In this sense,

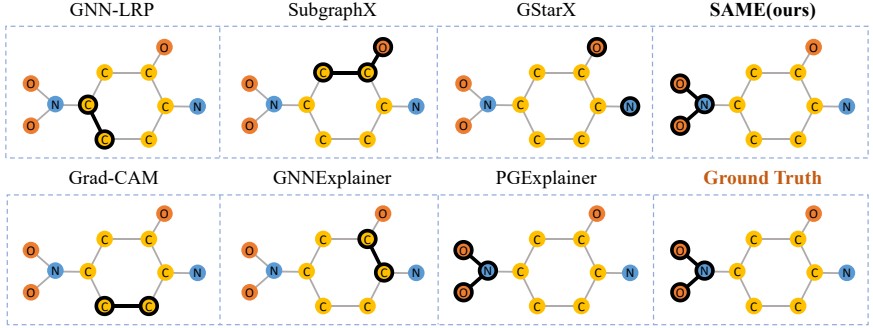

Figure 4: Comparison of the explanations on MUTAG with GIN classifier.

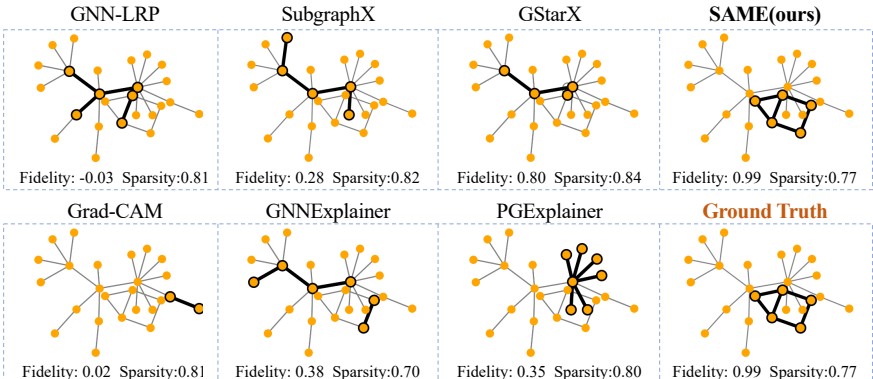

Figure 5: Comparison of the explanations on BA-2Motifs with GCN classifier.

SAME can better capture the adjectives-or-adverbs-like graph structures than other baselines. For instance, SubgraphX focuses on adjectives and adverbs but fails to capture the "but" word which bears significant weight under the contrasting relationship. Intuitively, without the "but", the contribution of "tortured" and "unsettling" should be negative. GStarX achieves to identify words that are consistent with the opinion such as "carefully" and "alive," yet overlooks the crucial transitional relationship between "alive" and "unsettling". Figure 4 shows the visualization of explanations on molecular graphs. The ground truth explanations (*i.e.*, functional group $-\mathbf{NO_2}$) of MUTAG are labelled by human experts. We see that SAME is able to provide the explanations the same as the ground truth. We also illustrate the explanation of the synthetic graph in Figure 5. The ground-truth label of all the graphs in BA-2Motifs is a 5-node-house-structure motif. Results show that SAME exactly finds the ground-truth explanation compared to other baselines. We leave more comparisons in Appendix G.2.

## 6    Conclusion

Structure-aware Shapley-based Multipiece Explanation provides strong explainability over GNN models, while this ability is limited by only using the single connected substructure. Moving forward from the theoretical deduction, we propose the SAME method for explaining GNN, a novel perturbation-based method that is aware of input graph structure, feature interactions, and multi-grained importance. Experimental results demonstrate that our SAME consistently outperforms SOTA methods on multiple benchmarks by a large margin with various metrics and provides a more understandable explanation.

**Limitations.** In the implementation of SAME, the Shapley value is obtained through approximation following [38]. Under this approximation, the fairness axioms discussed in Section 3.1 no longer hold. This is also identified as an unsolved issue for the Shapley value in machine learning by [23]. In addition, the scalability of SAME on large graphs can also become a potential challenge. As the time complexity shown in Section 4, the time overhead caused by the increase in the number of nodes will become very expensive when scaling up the size of the input graph.

## Acknowledgements

This work was funded in part by the National Key R&D Program of China (2021YFF1200804), National Natural Science Foundation of China (62001205), Shenzhen Science and Technology Innovation Committee (2022410129, KCXFZ2020122117340001), Shenzhen-Hong Kong-Macao Science and Technology Innovation Project (SGDX2020110309280100), Guangdong Provincial Key Laboratory of Advanced Biomaterials (2022B1212010003).

We would like to extend our sincere appreciation to Dr. Jialin Liu for her insightful suggestions on the manuscript composition and the elucidation of definitions.

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

# Appendix to "SAME: Uncovering GNN Black Box with Structure-aware Shapley-based Multipiece Explanation"

## Contents

This appendix comprises both theoretical and experimental materials and is structured as follows. Section A presents a notation table that includes all notations used in this paper. Section B provides the full theoretical analysis. Section C presents detailed algorithms related to Section 3 in the manuscript. Section D-F show the detailed settings of the experiments in this work. Section G provides additional quantitative and qualitative results.

## A   Notations

Table S1 contains the notations organized into three sections, each pertaining to different aspects of our paper. These include:

- **Graph**: These symbols are related to graph theory, representing elements such as graph nodes, edges, and other attributes of graphs.

- **MCTS**: These symbols represent elements related to the Monte Carlo Tree Search (MCTS) algorithm, such as MCTS nodes, actions, and rewards.

- **Importance score**: These symbols are used to denote values associated with measuring the importance of explanations in our explanation algorithm.

Table S1: Notations of the graph (see top part), MCTS (see middle part) and importance score (see bottom part).

| Notation | Description |
|---|---|
| $\mathcal{G}$ | The graph set to be explained. For any input graph $G \in \mathcal{G}$, $G = (V, X, E)$, $V = \{v_1, v_2, \ldots, v_n\}$ denotes node set, $E \in \mathbb{R}^{n \times n}$ denotes edge set and $X \in \mathbb{R}^{n \times d}$ denotes node feature set. |
| $\mathcal{G}_{ex}$ | The set of the explanation candidates for a given graph $G$. $\mathcal{G}_{ex} = \{G_{ex}^i\}_{i=1}^m$, where $G_{ex}^i \subseteq G$, $m$ is the number of explanation candidates |
| $G_{sub}^i$ | Each $G_{sub}^i$ is a connected component, and is called a substructure |
| $G_{com}^i$ | Each composite $G_{sub}^i$ contains more than two nodes or substructures. |
| $G_{ex}^i$ | Each $G_{ex}^i$ refers to a possible final explanation, $i.e.$, $G_{ex}^i = \{G_{sub_j}^i\}_{j=1}^{l_i}$, where $l_i$ is the number of substructures in explanation $G_{ex}^i$ |
| $T_i$ | The number of visit for node $i$ in MCTS |
| $R_i$ | The reward of node $i$ in MCTS |
| $C_i$ | The child set of node $i$ in MCTS |
| $A_i$ | A set of actions of MCTS node $i$ to reach its children |
| $W_i$ | The reward of all children for MCTS node $i$ |
| $\mathcal{N}$ | The node set of the corresponding MCTS of the input graph G. Each node of MCTS can be denoted as $N_i \in \mathcal{N}$, where $N_i$ is a structure containing $\{G_{ex}^i, T_i, R_i, A_i, C_i, W_i\}$. |
| $\pi$ | The strategy to choose the best child in MCTS. |
| $P_i$ | A set of players (including all nodes in the input $G$). |
| $f(\cdot)$ | A well-trained GNN model. |
| $I(\cdot, \cdot, \cdot)$ | The importance score, where we employ Shapley value here in this paper. |
| $|\cdot|$ | The size of a set. |

# B   Theoretical Analysis

## B.1   Definitions of Desire GNN Explanation Technique Properties

The following content is a detailed definition of desired properties in Table 1 of the main manuscript.

- **Graph-level tasks**: the GNN explanation method can handle the graph classification/regression tasks.

- **Node-level tasks**: the GNN explanation method can handle the node classification/regression tasks.

- **Feature interactions**: given a graph $G$ to be explained, when measuring the importance of the explanation result $G_{ex}$, the GNN explanation method can consider the influence of $\{G/G_{ex}\}$ on the importance of $G_{ex}$.

- **Structure awareness**: given a graph $G$ to be explained, when measuring the importance of the explanation result $G_{ex}$, the GNN explanation method is sensitive to the topology of the given input graph $G$.

- **Multipiece explanation**: the GNN explanation method can provide the explanation $G_{ex}$ that can be composed of one or more connected components.

- **Node-wise importance**: given any node $v_i \in G$, its importance $I(f(\cdot), v_i, G)$ can be considered by the GNN explanation method.

- **Substructure-wise importance**: given any substructure $G_{sub}^i \subseteq G$, it can be directly calculated as a whole through the GNN explanation method to obtain its importance $I(f(\cdot), G_{sub}^i, G)$. Note that any substructure is a connected component which has more than one node.

- **Composite-wise importance**: given any composite $G_{com}^i \subseteq G$ consisting of more than two nodes or substructures, it can be directly calculated as a whole through the GNN explanation method to obtain its importance $I(f(\cdot), G_{com}^i, G)$.

- **Priority-based integration**: given an explanation $G_{ex}^j \subseteq G$ with any size, the node $v_i \in \{G/G_{ex}^j\}$ will be added by the GNN explanation method on $G_{ex}^j$ to get a new explanation, if and only if for any node $v_k \in \{G/G_{ex}^j\}$ except $v_i$, $I(f(\cdot), \{v_i \cup G_{ex}^j\}, G) \geq I(f(\cdot), \{v_k \cup G_{ex}^j\}, G)$ always holds.

- **Redundancy consideration**: given an explanation $G_{ex}^j \subseteq G$, if there exist a node $v_i \in \{G/G_{ex}^j\}$ that satisfies $I(f(\cdot), \{G_{ex}^j/v_i\}, G) \geq I(f(\cdot), G_{ex}^j, G)$, the GNN explanation technique would not choose $G_{ex}^j$ as the final explanation.

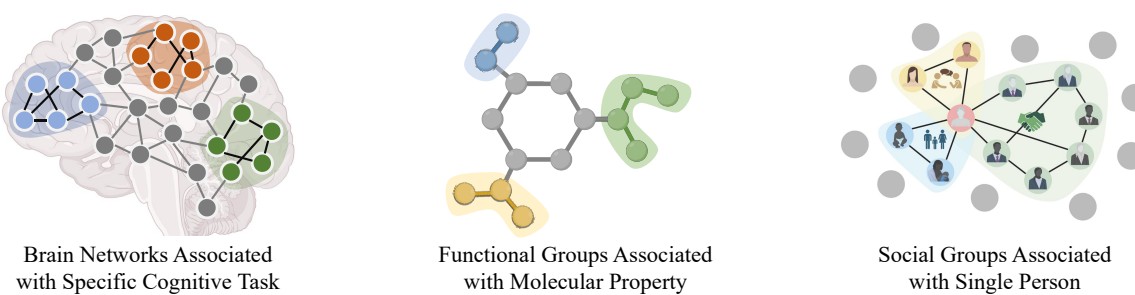

| Brain Networks Associated with Specific Cognitive Task | Functional Groups Associated with Molecular Property | Social Groups Associated with Single Person |

Figure S1: Examples of the characteristics and properties within a graph or node.

As Figure S1 shows, the characteristics and properties within a graph or node tend to be *jointly* influenced by more than one high-order connected community of the graph. The "multipiece explanation" property is aware of such an attribute by considering whether multiple disconnected pieces are allowed to appear together in the explanation output. For example, when "A" and "B" are two orthogonal connected components and the ground truth explanation for a prediction is "A and B", then the GNN explanation method has a probability of providing the correct "A and B" explanation.

We call a GNN explanation method that has the "structure-awareness" property if it can process the structural information of the input graph, whether through explicit mechanisms, as demonstrated by GstarX, or implicit modalities, as exemplified by SubgraphX and this study.

## B.2 Discussion about Four Fairness Properties

*Efficiency* demonstrates that the aggregated importance of individual substructures is equivalent to the prediction of GNN on the entire substructure set. *Symmetry* and *Dummy* take into account the equal importance of substructures with the same interactions and the substructures that are completely unimportant, respectively. *Monotonicity* ensures that the explanation method is consistent in the trend of results between different well-trained GNN models given the same dataset.

## B.3 Detailed Proof of Regret Bound

The proper definition of the regret phenomenon that the algorithm or the function $f$ encounters must be considered to make the objective clear. Prior studies have recognized two principal definitions:

We recall the Lemma 1 in the main manuscript as follows.

**Lemma 1.** *Let $d$ be the $\nu$-near-optimality dimension (where $\nu$ is defined in Assumption 3 of the main manuscript), and $C$ is the corresponding constant. Then $|I_h| \leq C\delta(h)^{-d}$.*

*Proof.* From Assumption 3 in the main manuscript, each cell $(h, i)$ contains a ball of radius $\nu\delta(h)$ centered at $x_{h,i}$. Consequently, if $|I_h| = |\{x_{h,i} \in X_{\delta(h)}\}|$ exceeds $C\delta(h)^{-d}$, this would imply the existence of more than $C\delta(h)^{-d}$ disjoint $l$-balls of radius $\nu\delta(h)$, each centered within $X_{\delta(h)}$. This assertion would contradict the definition of $d$. $\qquad\square$

We recall the Theorem 1 in the main manuscript as follows.

**Theorem 1.** *Let us write $h(n)$ the smallest integer $h$ such that $C\sum_{l=0}^{h}\delta(l)^{-d} \geq n$. Then the loss of algorithm is bounded as $r_n \leq \delta(h(n))$.*

*Proof.* Consider the tuple $(h_{\max}, j_{\max})$ to be the node with the maximum depth that the algorithm expands up to the $n$th round. Given that the algorithm's expansion is limited to nodes within the set $I$, we can provide that the total number of nodes expanded, denoted by $n$, adheres to the following condition:

$$n = \sum_{l=0}^{h_{\max}} K^l \sum_{j=0}^{K^l-1} 1\!\!1\{(h,j) \text{ has been expanded}\}$$

$$\leq \sum_{l=0}^{h_{\max}} |I_l|$$

$$\leq C \sum_{l=0}^{h_{\max}} \delta(l)^{-d},$$

from Lemma 1. Now from the definition of $h(n)$ we have $h_{\max} \geq h(n)$. Finally, since node $(h_{\max}, j_{\max})$ has been expanded, we have that $(h_{\max}, j_{\max}) \in I$, thus $f(x(n)) \geq f(x_{h_{\max},j_{\max}}) \geq f^\star - \delta(h_{\max}) \geq f^\star - \delta(h(n))$. $\square$

**Corollary 1.** *Assume that $\delta(h) = c\gamma^h$ for some constants $c > 0$ and $\gamma < 1$, if $d > 0$, then the loss decreases polynomially fast:*

$$r_n \leq \left(\frac{C}{1-\gamma^d}\right)^{1/d} n^{-1/d}$$

*If $d = 0$, then the loss decreases exponentially fast:*

$$r_n \leq c\gamma^{n/C-1}$$

*Proof.* From Theorem 1, whenever $d > 0$ we have

$$n \leq C \sum_{l=0}^{h(n)} \delta(l)^{-d} = Cc^{-d} \frac{\gamma^{-d(h(n)+1)-1}}{\gamma^{-d}-1} \tag{1}$$

thus $\gamma^{-dh(n)} \geq \frac{n}{Cc^{-d}}(1-\gamma^d)$, therefore we can also have:

$$r_n \leq \delta(h(n)) \leq c\gamma^{h(n)} \leq \left(\frac{C}{1-\gamma^d}\right)^{1/d} n^{-1/d} \tag{2}$$

Now, if $d = 0$ then $n \leq C\sum_{l=0}^{h(n)} \delta(l)^{-d} = C(h(n)+1)$, and we deduce that the loss is bounded as $r_n \leq \delta(h(n)) = c\gamma^{n/C-1}$. $\square$

## B.4 Analysis of Time Complexity

Given a graph $G = (V, E)$, we take a single iteration as an example to analyze the time complexity of the expansion-based MCTS in our proposed SAME. A single iteration includes four steps: selection, expansion, simulation and backpropagation [1].

### B.4.1 Important Substructure Initialization Phase

In this phase, the goal of our SAME is to figure out a group of important substructures whose size is smaller than $\gamma$. In the selection step, our method chooses an unvisited node from the graph or chooses the node with the highest reward if all nodes have been visited ($\mathcal{O}(|V|)$). Then, in the expansion step, our method selects the node within 1-hop neighbors whose value is largest

according to $G$, which requires $\mathcal{O}(1)$, and adds it to the tree. In the `simulation step`, only the adjacency node will be appended each time. For the graph $G$, in the worst case SAME will append all the nodes by visiting all the edges in the graph. Thus, the time complexity of simulation is $\mathcal{O}(|V| + |E|)$. For the `backpropagation step`, since the maximum size of a substructure is $\gamma$, the depth of the MCTS tree will not exceed $\gamma$. The time complexity of reward backpropagation is bounded by $\mathcal{O}(\gamma)$. Since we perform $M_1$ iterations, according to the law of multiplication, the time complexity of the first phase is $\mathcal{O}(M_1) \times \mathcal{O}(selection) \times \mathcal{O}(expansion) \times \mathcal{O}(simulation) \times \mathcal{O}(backpropagation) = \mathcal{O}(M_1 \gamma |V|^2 + M_1 \gamma |V| \times |E|)$.

### B.4.2   Explanation exploration phase

In the `selection step`, the MCTS is initialized with an unvisited substructure from *Important Substructure Set* (size $= K$, where $K \ll |V|$) or chooses the substructure with the highest reward if all substructures have been visited ($\mathcal{O}(K)$). The `expansion step` is $\mathcal{O}(1)$ as we only need to append the substructure with the largest value to the current state. In the following `simulation step`, our method will append other substructures to the current state until the size exceeds the sparsity limit, which is bounded by $\mathcal{O}(2^K)$ for finding all possible combination. Notice that $\mathcal{O}(2^K)$ can be very large, we set a maximum simulation time $t_s$ as budget which requires $\mathcal{O}(t_s)$. Since the final explanation is highly related to the sparsity, the cost of `backpropagation step` is bounded by $\mathcal{O}(\frac{|V| \times (1 - sparsity)}{\gamma}))$, where $\gamma$ is the maximum size of each substructure. With $M_2$ iterations, the time complexity in this phase is $\mathcal{O}(M_2) \times \mathcal{O}(selection) \times \mathcal{O}(expansion) \times \mathcal{O}(simulation) \times \mathcal{O}(backpropagation) = \mathcal{O}(M_2 K t_s \frac{|V| \times (1 - sparsity)}{\gamma})$.

Overall, time complexity of SAME is $\mathcal{O}(M_1 \gamma |V|^2 + M_1 \gamma |V| \times |E| + M_2 K t_s \frac{|V| \times (1 - sparsity)}{\gamma})$. As the $t_s$, $\gamma$, $K$, $M_1$, $M_2$ and *sparsity* is predefined, therefore SAME is a polynomial-time method under these constraints.

## C   Detailed Pseudo Code of SAME

In this section, we are going to provide a detailed overview of the processes of SAME which includes *important substructure initialization* and *explanation exploration*.

Algorithm 1 outlines the process of using Monte Carlo Tree Search (MCTS) to find a set of important substructures, as mentioned in Section 3.1 of the main manuscript. Algorithm 2 and 3 illustrate the method to discover an optimal explanation from these important substructures, as described in Section 3.2. Algorithm 4 provides a detailed description of how the importance of an explanation is assessed using the structure-aware Shapley value, which was proposed in Section 2.1 of your document. This algorithm outlines the steps necessary to approximate the Shapley value for a given explanation, effectively measuring the contribution of each individual component via sampling.

## D   Dataset Description

We provide details of the datasets used in our experiments, including BBBP [11], MUTAG [2], BA-2Motifs [8], BA-Shapes [12], Graph-SST2 [7], and Graph-SST5 [7].

**Molecular graphs.**   We use the BBBP [11] containing approximately 2000 molecular graphs, which are classified into two classes over the property of blood–brain barrier penetration (BBBP). Another dataset MUTAG [2] is a collection of molecules with $-NO_2$ functional groups. The goal is to predict whether these molecules are mutation-induced.

**Barabási-Albert graphs.**   The BA-2Motifs [8] and the BA-Shapes [12] are used for graph classification and node classification respectively. For each instance in BA-2Motifs, it is a Barabási-Albert graph attached by motifs with a structure either house-like or five-node-cycle-like. The instances are labelled according to the type of motifs they get. The graph in BA-Shapes is a Barabási-Albert graph with house-structured network motifs. The nodes in the graph will be classified into four classes, with labels 0 for the nodes belonging to the original graph and labels 1-3 for the nodes on the middle, bottom, or top of the house-like structures respectively.

**Sentiment graphs.**   The sentiment graphs are built from real-world text sequences, and labelled according to the semantic meanings. Specifically, the nodes in the graph represent the words with an

---

**Algorithm 1:** Important Substructure Initialization

---

**Input:** $f(\cdot)$, well-trained GNN model; $G$, graph to be explained; $M_1$, max iteration number of MCTS; $\gamma_1$, threshold of max single substructure size; $\pi$, child selection strategy

**Output:** $\mathcal{N}$, all nodes in MCTS

**1** Initialize the root of the MCTS as $N_0$

**2** $G_{sub}^0 \leftarrow \emptyset$

**3** $\mathcal{N} \leftarrow \{N_0\}$

**4 for** $i = 1, 2, \ldots, M_1$ **do**

**5**     $N_{cur} \leftarrow N_0$

**6**     $curPath \leftarrow [N_0]$

**7**     $G_{sub}^{cur} \leftarrow G_{sub}^0$

**8**     **while** $|G_{sub}^{cur}| < N_{max}$ **do**

**9**        **forall** $v_j$ *in the* $\{adj(G_{sub}^{cur})\}$ **do**

**10**           Expand $N_i$ to get $N_j$

**11**           $G_{sub}^j \leftarrow G_{sub}^i \cup \{v_j\}$

             // Compute the Shapley-based reward according to Algorithm 4

**12**           $R_j \leftarrow I(f(\cdot), G_{sub}^j, G)$

**13**           $\mathcal{N} \leftarrow \mathcal{N} \cup \{N_j\}$

**14**        **end**

**15**        Select the child $N_{next}$ and its substructure $G_{sub}^{next}$ according to the strategy $\pi$

**16**        $N_{cur} \leftarrow N_{next}$

**17**        $G_{sub}^{cur} \leftarrow G_{sub}^{next}$

**18**        $curPath \leftarrow curPath + N_{next}.$

**19**        $\mathcal{N} \leftarrow \mathcal{N} \cup \{N_{cur}\}$

**20**     **end**

**21**     $N_l \leftarrow N_{cur}$ // $N_l$ is a leaf node

**22**     **forall** *node* $N_{path_i}$ *in the* $curPath$ **do**

**23**        $T_{path_i} \leftarrow T_{path_i} + 1$

**24**        $W_{path_i} \leftarrow W_{path_i} + I(f(\cdot), G_{sub}^l, G)$

**25**     **end**

**26 end**

---

---

**Algorithm 2:** Explanation Exploration

---

**Input:** $f(\cdot)$, well-trained GNN model; $G$, graph to be explained; $\mathcal{N}$, set of all the nodes in MCTS; $K$, number of the top most important structures; $K_t$, threshold to use MCTS; $M_2$, max iteration number of MCTS; $\gamma_2$, max explanation size; $\pi$, child selection strategy.

**Output:** $G_{ex}$, the best explanation.

**1** Sort the $\mathcal{N}$ in descending order of the corresponding reward $R_i$.

**2** $\mathcal{N}_K \leftarrow$ top $K$ substructures $\{N_1, N_2, \ldots, N_K\}$ in $\mathcal{N}$

**3** $\mathcal{G}_{ex} \leftarrow \{(G_{sub_i}, R_i)\}_{i=1}^{M_2}$

    // Find explanations through MCTS in Algorithm 3.

**4** $\mathcal{G}'_{ex} \leftarrow \text{MCTS}(f(\cdot), G, M_2, N_{max}, \pi, \mathcal{N}_K)$

**5** $\mathcal{G}_{ex} \leftarrow \mathcal{G}_{ex} \cup \mathcal{G}'_{ex}$

**6** Sort the $\mathcal{G}_{ex}$ in descending order of $R$.

    // $G_{ex}$ is the explanation at the first element in the sorted $\mathcal{G}_{ex}$

**7** $G_{ex} \leftarrow \mathcal{G}_{ex}[0]$

---

**Algorithm 3:** MCTS Explanation Exploration

**Input:** $f(\cdot)$, well-trained GNN model; $G$, graph to be explained; $M_2$, max iteration number of MCTS; $\gamma_2$, threshold of max explanation size; $\pi$, child selection strategy, $\mathcal{N}_K$, the substructure set.

**Output:** $\mathcal{G}_{ex}$, the explanation set found in MCTS

1 Initialize the root of the MCTS as $N_0'$
2 $G_{ex}^0 \leftarrow \emptyset, \mathcal{G}_{ex} \leftarrow \{G_{ex}^0\}$
3 **for** $i = 1, 2, \ldots, M_2$ **do**
4     $N_{cur}' \leftarrow N_0', curPath \leftarrow [N_0'], G_{ex}^{cur} \leftarrow G_{ex}^0$
5     **while** $|G_{ex}^{cur}| < \gamma_2$ **do**
6        **forall** $N_j$ *in the* $\{\mathcal{N}_K \backslash N_{cur}'\}$ **do**
          `// Expand` $N_{cur}'$ `with` $N_j$ `to get` $N_j'$`:`
7           $N_j' \leftarrow N_{cur}' \cup N_j$
8           $G_{ex}^j \leftarrow G_{ex}^i \cup G_{sub_j}$
          `// Compute reward according to Algorithm` 4
9           $R_j' \leftarrow I(f(\cdot), G_{ex}^j, G)$
10        **end**
11        Select the child $N_{next}'$ and its substructure $G_{ex}^{next}$ according to the strategy $\pi$
12        $N_{cur}' \leftarrow N_{next}', G_{ex}^{cur} \leftarrow G_{ex}^{next}$
13        $curPath \leftarrow curPath + N_{next}'$.
14     **end**
15     $N_l' \leftarrow N_{cur}'$ `//` $N_l'$ `is a leaf node`
16     $\mathcal{G}_{ex} \leftarrow \mathcal{G}_{ex} \cup \{(G_{ex}^{cur}, R_{cur}')\}$
17     **forall** *node* $N_{path_i}$ *in the* $curPath$ **do**
18        $T_{path_i}' \leftarrow T_{path_i}' + 1$
19        $W_{path_i}' \leftarrow W_{path_i}' + I(f(\cdot), G_{ex}^l, G)$
20     **end**
21 **end**

---

**Algorithm 4:** Importance Scoring via Shapley Value

**Input:** $f(\cdot)$, well-trained GNN model; $G$, graph data; $G_{ex}^i$, explanation of $G$; $T$, sampling times; $k$, number of neighboring hop

**Output:** $\mathcal{I}_i$, importance of $G_{ex}^i$ for GNN $f(\cdot)$.

1 Obtain the $k$-hop neighbor nodes of substructure $G_{ex}^i$, denoted as $\{v_{n_i+1}, v_{n_i+2}, \ldots, v_{n_i+k_i}\}$.
2 Then, the players set $P_{i,khop} = \{G_{ex}^i, v_{n_i+1}, v_{n_i+2}, \ldots, v_{n_i+k_i}\}$
3 **for** $i = 1, 2, \ldots, T$ **do**
4     Randomly sample a set $S_i \subseteq \{P \backslash G_{ex}^i\}$
5     Obtain $f(S_i \cup \{G_{ex}^i\})$ and $f(S_i)$ by setting the features of the nodes not in the input structure with zero.
    `//` $m(S_i, G_{ex}^i)$ `denotes the marginalized contribution`
6     $m(S_i, G_{ex}^i) \leftarrow f(S_i \cup \{G_{ex}^i\}) - f(S_i)$.
7 **end**
8 $I(f(\cdot), G_{ex}^i, G) \leftarrow \frac{1}{T} \sum_{t=1}^T m(S_i, G_{ex}^i)$
9 $\mathcal{I}_i \leftarrow I(f(\cdot), G_{ex}^i, G)$

initial embedding from the pre-trained BERT [3] and the edges denote the relationships between the words which are extracted by the Biaffine parser [5]. We take experiments on the Graph-SST2 and Graph-SST5 datasets for the sentiment graphs. Note that both Graph-SST2 and Graph-SST5 are the datasets for graph classification of two classes and five classes respectively.

# E  Metrics

The Fidelity, Inverse Fidelity, and Sparsity are formally defined as:

$$\text{Fidelity}(G, G_{ex}) = [f(G)]_{c^*} - [f(G/G_{ex})]_{c^*} \tag{3}$$

$$\text{Fidelity}_{\text{Inv}}(G, G_{ex}) = [f(G)]_{c^*} - [f(G_{ex})] \tag{4}$$

$$\text{Sparsity}(G, G_{ex}) = 1 - \frac{|G_{ex}|}{|G|} \tag{5}$$

Harmonic Fidelity (H-Fidelity) takes a harmonic mean of normalized fidelity (N-Fidelity) and normalized inverse fidelity (N-Fidelity$_{\text{Inv}}$). According to [15], the H-Fidelity can be formally defined as:

$$\text{N-Fidelity}(G, G_{ex}) : m_1 = \text{Fidelity}(G, G_{ex}) \cdot \text{Sparsity}(G, G_{ex}) \tag{6}$$

$$\text{N-Fidelity}_{\text{Inv}}(G, G_{ex}) : m_2 = \text{Fidelity}_{\text{Inv}}(G, G_{ex}) \cdot (1 - \text{Sparsity}(G, G_{ex})) \tag{7}$$

$$\text{H-Fidelity}(G, G_{ex}) = \frac{(1 + m_1)(1 - m_2)}{(2 + m_1 - m_2)} \tag{8}$$

# F  Detailed Experimental Settings

In this section, we provide the computational details and detailed hyperparameter settings on different datasets.

## F.1  Computation Details

All experiments were performed on a single Nvidia V100 GPU with an Intel Xeon Gold 5218 CPU. Since it takes more than 24 hours for some baselines to explain the whole dataset, our inference time analysis is implemented by randomly sampling from the datasets. For BBBP, BA-2Motifs and BA-Shapes, we use all the test data. And all the data (train, evaluation and test data) is used for MUTAG dataset. For semantic graphs like Graph-SST2 and Graph-SST5, we randomly choose 30 graphs.

## F.2  Hyperparameter Settings

We adhere to the hyperparameter settings for GNN training as described in [13, 15], detailed in Table S2. For the qualitative analysis, the GCN in Figures 2, 4, S2, and S3 aligns with the settings depicted in Table S2. Moreover, the GIN featured in Figure 3 is trained for 800 epochs with a batch size of 64 and without adding self-loops, while all other hyperparameters remain constant to Table S2.

Table S2: Hyperparameters for training the GCN models on different datasets. All quantitative results are verified under the following conditions.

| Hyperparameter | BBBP | MUTAG | Graph-SST2 | Graph-SST5 | BA-2Motifs | BA-Shapes |
|---|---|---|---|---|---|---|
| # Layers | 3 | 3 | 3 | 3 | 3 | 3 |
| Hidden dimensions | $[128, 128, 128]$ | $[128, 128, 128]$ | $[128, 128, 128]$ | $[128, 128, 128]$ | $[20, 20, 20]$ | $[20, 20, 20]$ |
| Dropout | 0.0 | 0.0 | 0.0 | 0.0 | 0.0 | 0.0 |
| Readout method | max | mean | max | max | mean | identity |
| Learning rate | 0.001 | 0.005 | 0.001 | 0.001 | 0.001 | 0.05 |
| Batch size | 32 | 32 | 128 | 128 | 64 | 1 |
| # Epochs | 200 | 1000 | 50 | 50 | 800 | 400 |
| Weight decay | $5 \times 10^{-4}$ | 0 | 0 | 0 | 0 | $5 \times 10^{-4}$ |
| Add self-loop | Yes | Yes | Yes | Yes | Yes | Yes |

Table S3: The quality of the explanations from SAME with different hyperparameters.

| $\gamma\backslash K$ | 1 | 3 | 5 | 7 | 9 |
|---|---|---|---|---|---|
| 1 | 0.2511 | 0.3319 | 0.3548 | 0.3938 | 0.4026 |
| 2 | 0.2851 | 0.3121 | 0.3418 | 0.3652 | 0.3786 |
| 3 | 0.2938 | 0.3180 | 0.3297 | 0.3637 | 0.3741 |
| 4 | 0.3134 | 0.3355 | 0.3491 | 0.3610 | 0.3540 |
| 5 | 0.3134 | 0.3387 | 0.3504 | 0.3591 | 0.3598 |

Table S4: The average inference time (seconds) per sample under different hyperparameters.

| $\gamma\backslash K$ | 1 | 3 | 5 | 7 | 9 |
|---|---|---|---|---|---|
| 1 | 0.0990 | 0.3300 | 1.289 | 5.178 | 15.95 |
| 2 | 0.0889 | 0.3129 | 1.137 | 3.817 | 11.88 |
| 3 | 0.0874 | 0.3986 | 0.8344 | 2.746 | 7.047 |
| 4 | 0.0932 | 0.2846 | 0.8483 | 2.296 | 5.330 |
| 5 | 0.0914 | 0.2822 | 0.8427 | 2.191 | 5.796 |

The hyperparameters in SAME include $\gamma$ and $K$ as we didn't use a time budget in our MCTS. In order to see the effect of these two hyperparameters, we run our SAME by randomly selecting 30 graphs from Graph-SST5 datasets using GCN, the fidelity w.r.t. $\gamma$ and $K$ are shown in the Table S3, and we report the average inference time in Table S4. Notice that since we only selected 30 graphs from Graph-SST5 to obtain the results, the results provided in Table S3 might be slightly different from the results given in the main manuscript.

It is noteworthy that $\gamma$=1 could lead to optimal fidelity in most benchmarks. However, when $\gamma$ is small, the final explanation may be composed of disconnected nodes at different positions in the graph, which usually causes the visualization of explanation results not human-understandable. Therefore, we fine-tuned $\gamma$ and $K$ according to the visualization of explanation results so that the provided explanations are more human-intuitive. For the inference time, a larger $K$ leads to a larger search space for the *explanation exploration phase*. When $\gamma$ is small, the size of a single candidate substructure is small, indicating that we need to select more substructures from the candidates to obtain an explanation with the desired sparsity. Thus, the inference time increases as the depth of searching grows. The detailed hyperparameter settings are provided in Table S5. For the sentiment network Twitter [13] and molecular network BACE [11], we use them to compare SAME with more SOTA explainers in Section G.3

Table S5: The hyperparameters of SAME for different datasets.

| Hyperparameter | BBBP | MUTAG | Graph-SST2 | Graph-SST5 | BA-2Motifs | BA-Shapes | Twitter | BACE |
|---|---|---|---|---|---|---|---|---|
| $\gamma$ | 5 | 2 | 3 | 3 | 3 | 5 | 3 | 2 |
| $K$ | 7 | 7 | 7 | 7 | 7 | 7 | 7 | 7 |

# G   Additional Results

## G.1   Results under Other Metrics

Here, we present the results over inverse fidelity and harmonic fidelity under the same experimental settings in Table S2. Table S6 demonstrates that SAME is competitive over the 5/6 datasets. Table S7 reports the harmonic fidelity quantities. SAME outperforms the baselines in all the graph tasks.

## G.2   Additional Visualization Results

We are presenting additional visualization results for BBBP and Graph-SST2 datasets. Figure S2 presents explanations for the BBBP dataset. SAME identifies critical functional groups (for instance, carbonyl group =C=O), which result in a high fidelity score. Figure S3 provides a visualization for the negative label of the Graph-SST2 dataset. In this case, SAME uniquely identifies the word "only," which contributes to the negative prediction of the GNN due to the contrasting relationship it establishes.

Table S6: Comparison of our SAME and other baseline explainers using 1 - Inverse Fidelity.

| Dataset | Graph tasks | | | | | Node tasks |
|---|---|---|---|---|---|---|
| | Molecular graph | | Semantic graph | | Synthetic graph | |
| Methods | BBBP | MUTAG | Graph-SST2 | Graph-SST5 | BA-2Motifs | BA-Shapes |
| Grad-CAM [9] | 0.897±0.029 | 0.901±0.048 | 1.006±0.029 | 0.937±0.047 | 0.587±0.092 | - |
| GNNExplainer [12] | 0.829±0.012 | 0.810±0.007 | 0.855±0.037 | 0.860±0.034 | 0.562±0.015 | 0.953±0.032 |
| PGExplainer [8] | 0.838±0.036 | 0.793±0.030 | 0.956±0.036 | 0.850±0.058 | 0.582±0.021 | 0.902±0.008 |
| GNN-LRP [10] | 0.670±0.044 | 0.768±0.013 | 0.784±0.047 | 0.771±0.041 | 0.560±0.015 | **0.988±0.010** |
| SubgraphX [14] | **1.010±0.011** | 1.053±0.017 | 1.008±0.008 | 1.022±0.040 | **0.906±0.162** | 0.978±0.034 |
| GStarX [15] | 0.939±0.011 | **1.124±0.025** | **1.072±0.003** | **1.140±0.031** | 0.587±0.117 | - |
| **SAME** | 0.993±0.014 | 1.095±0.008 | 1.058±0.010 | 1.039±0.001 | 0.857±0.046 | 0.881±0.000 |

Note: The previous SOTA results on different datasets are marked with an underline. *Relative Improve* denotes the relative improvement of our SAME method over the SOTA methods.

Table S7: Comparison of our SAME and other baseline explainers using Harmonic Fidelity.

| Dataset | Graph tasks | | | | | Node tasks |
|---|---|---|---|---|---|---|
| | Molecular graph | | Semantic graph | | Synthetic graph | |
| Methods | BBBP | MUTAG | Graph-SST2 | Graph-SST5 | BA-2Motifs | BA-Shapes |
| Grad-CAM [9] | 0.520±0.002 | 0.527±0.002 | 0.538±0.005 | 0.529±0.003 | 0.502±0.014 | - |
| GNNExplainer [12] | 0.504±0.002 | 0.505±0.007 | 0.507±0.001 | 0.512±0.001 | 0.487±0.027 | 0.514±0.001 |
| PGExplainer [8] | 0.511±0.002 | 0.500±0.006 | 0.528±0.003 | 0.516±0.000 | 0.493±0.023 | 0.504±0.003 |
| GNN-LRP [10] | 0.488±0.003 | 0.501±0.013 | 0.495±0.001 | 0.495±0.002 | 0.497±0.027 | 0.518±0.002 |
| SubgraphX [14] | 0.560±0.002 | 0.552±0.002 | 0.528±0.004 | 0.541±0.003 | 0.548±0.007 | **0.528±0.003** |
| GStarX [15] | 0.510±0.002 | 0.599±0.011 | 0.533±0.005 | 0.542±0.005 | 0.506±0.016 | - |
| **SAME** | **0.571±0.001** | **0.604±0.022** | **0.540±0.021** | **0.566±0.000** | **0.554±0.007** | 0.517±0.000 |
| **Relative Improve** | **1.96%↑** | **0.83%↑** | **1.31%↑** | **4.43%↑** | **1.09%↑** | - |

Note: The previous SOTA results on different datasets are marked with an underline. *Relative Improve* denotes the relative improvement of our SAME method over the SOTA methods.

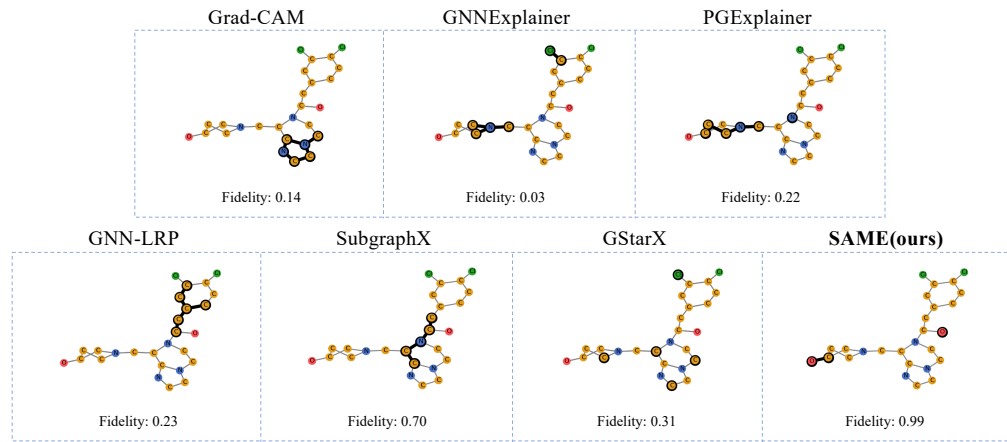

Figure S2: Comparison of the explanations on BBBP with GCN classifier.

Sentence: "shocking only in that it reveals the filmmaker's bottomless pit of self - absorption."

Figure S3: Comparison of the explanations on Graph-SST2 with GCN classifier for a negative prediction.

## G.3 Additional Quantitative Analysis

Here we report more quantitative analysis with alternative SOTA methods and datasets. We tested SAME on GAT and GIN, and compared SAME with other SOTA algorithms (GraphSVX [4] and OrphicX [6]). In table S8, We report the averaged fidelity over a sparsity of $[0.5, 0.6, 0.7, 0.8]$. For the inference time plus fidelity, they are summarized in Table S9. SAME not only surpasses the benchmarks' performance by alternative GNN architectures, but it also exhibits superior efficiency in elucidating explanations.

Table S8: Comparison of SAME and other baseline explainers with different GNN architecture.

| Dataset | GAT | | GIN | | GCN | |
|---|---|---|---|---|---|---|
| Methods | Graph-SST2 | Graph-SST5 | Graph-SST2 | Graph-SST5 | Twitter | BACE |
| GraphSVX [4] | -0.0136 | -0.0108 | 0.0839 | 0.161 | 0.0691 | 0.2061 |
| OrphicX [6] | 0.071 | 0.104 | 0.258 | 0.176 | 0.1701 | 0.3277 |
| SubgraphX [14] | 0.060 | 0.060 | 0.222 | 0.254 | _0.2903_ | _0.3906_ |
| GStarX [15] | _0.134_ | _0.114_ | _0.274_ | _0.263_ | 0.2501 | 0.3903 |
| **SAME** | **0.142** | **0.222** | **0.400** | **0.264** | **0.3127** | **0.7154** |

Note: The previous SOTA results on different datasets are marked with an underline.

Table S9: Comparison of SAME and other benchmarks in both fidelity and inference time (per sample). The architecture of our model is GCN.

| Metric | Methods | GraphSVX [4] | OrphicX [6] | SubgraphX [14] | GStarX [15] | **SAME** |
|---|---|---|---|---|---|---|
| Fidelity | Twitter | 0.0691 | 0.1701 | _0.2903_ | 0.2501 | **0.3127** |
| | BACE | 0.2061 | 0.3277 | _0.3906_ | _0.3903_ | **0.7154** |
| Time | Twitter | 0.0885 | 0.4942 | 197.1 | 36.26 | 25.85 |
| | BACE | 0.0915 | 0.2361 | 44.41 | 48.81 | 26.98 |

Note: The previous SOTA results on different datasets are marked with an underline.

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
