# OpenReview forum: "SAME: Uncovering GNN Black Box with Structure-aware Shapley-based Multipiece Explanations"
_NeurIPS.cc/2023/Conference — NeurIPS 2023 poster_

### Official Review · Reviewer_xcjE · 2023-07-05

**Soundness:** 2 fair
**Presentation:** 2 fair
**Contribution:** 3 good
**Rating:** 6
**Confidence:** 3

**Summary:**

A GNN-based Shapley value for GNN explanation is proposed, which is a novel Structure-Aware Shapley-based Multi-Explanation (SAME) technique for fairly considering the multi-level structure-aware feature interactions over an input graph It is computed by an expansion-based Monte Carlo tree search (MCTS) and its main advantage over existing methods is its theoretical foundation.
Experiments on multiple datasets also highlight practical benefits.


**Strengths:**

- **Clarity:**
Table 1 clearly points out the differences/additional features in comparison to related methods. The figures have been designed with care and are visually appealing. In particular, the appendix contains some beautiful figures. However, some arguments could be made more clear (see weaknesses). Some larger parts of the paper are not well understandable because information is missing (for details see points of minor critique).

- **Originality:**
Table 1 explains the novelty of SAME relative to competing methods. While all of the listed features are realised by some methods, SAME is able to offer their combination.
The authors claim that another distinguishing feature is a theoretical foundation. (However, other Shapley based measures are also benefit from a well established related theory.)
The proposed method SAME is quite similar to SubgraphX. Primarily, the pruning-based algorithm is replaced by an expansion-based algorithm. Thus, the main idea and game theoretic framework are not novel.

- **Quality:**
The figures are well designed and the text is well structured. The experiments seem standard. The main algorithmic contributions are difficult to understand, however.


- **Significance:**
Experiments suggest a clear improvement over competing methods on multiple datasets.
Only some results on synthetic graphs in the appendix are not competitive.


**Weaknesses:**

- The biggest issue is that considerable parts of the paper are not well understandable. The algorithm and the contribution of its parts are not clear. Neither the game theoretic approach is explained. It would also be important to discuss the detailed differences between SAME and the similar approach SubgraphX.

- The Shapley value is only computed approximately (as common).

- The search space cannot be covered completely. Thus, the algorithm has therefore no guarantees to find a global optimum (as common).

- The implications of the Theorem 1 need to be explained better. Currently, it does not seem to provide clear practical value, since the fact that one bound might be better than another one does not imply that the corresponding loss is also smaller..

- Missing literature?
GraphSVX: Shapley Value Explanations for Graph Neural Networks. ECML PKDD 2021.

- **Reproducibility:** No code has been submitted.

**Points of minor critique:**

- The rules of the underlying multiplayer game are not explained.

- Lines 110-121, which are supposed to explain the main contribution (the algorithm that explores the search space), are not really understandable. At least a correction of the grammar would be helpful.

- Lines 130-134: It is impossible to follow the argument why the proposed algorithm requires fewer smoothness constraints on the function $f$, since the pruning algorithm is not introduced before.
It might help to move the discussion of related work to the beginning of the paper.

- Figure 1 is not sufficient to explain the overall approach. The main components are not explained in the text.


**Questions:**

For what kind of GNNs does the proposed explainer work? Could attention be considered?


**Limitations:**

The limitations have been pointed out under weaknesses and were mostly discussed by the authors. I do not foresee a negative societal impact of this work.

---

> ### Author Rebuttal · Authors · 2023-08-10
>
> Thank you for your careful review and comments. Below please find our point-to-point responses to your comments.
>
> **Q1. The algorithm and the contribution of its parts are not clear. Neither the game theoretic approach is explained. It would also be important to discuss the detailed differences between SAME and the similar approach SubgraphX.**
>
> 1. Detailed algorithm of SAME: Please kindly find the detailed algorithm of our method provided in **Algorithm 1-4** in the Supplementary Material.
>
> 2. Contribution: In short, the key contribution of our work includes (1) *Theoretical aspect:*  i) We summarize and refine the characteristics of previous work and formally provide several desired properties (see Table 1 in the main manuscript) that can be considered by explanation methods for GNNs. ii) We provide the loss bound of the MCTS-based explanation techniques and further prove the superiority of our expansion-based MCTS in SAME compared to the previous work. (2) *Empirical aspect:* Our experiments cover both real-world and synthetic datasets. The results reveal that i) SAME consistently and significantly outperforms previous SOTA with different metrics under the same condition. ii) SAME qualitatively achieves a more human-intuitive explanation compared to previous methods across multiple datasets.
>
>    Our work is related to the SubgraphX, but it is different from it because our approach considers multiple explanations, redundancy in explanation, and node-wise importance of the given graph, simultaneously (see Table 1 in the main manuscript). Our approach is based on MCTS, but with a novel two-phase expansion-based MCTS version, which is shown to outperform the subgraphX, and outperform or be competitive with other SOTA methods in both inference time and explainability (see Table 2 and Table 3 in the main manuscript; Table S3 and Table S4 in the supplementary material; *Overall response: Q4* for additional experiments for detailed results).
>
> 3. Game theoretic approach: We have revised the Preliminaries in Section 2 to make it more clear. In short, finding the explanation for a given graph $G$ by using an importance scoring function $I(f(\cdot),G_{ex},G)$ can be formalized as:
>    $$G_{ex}^* = \mathop{\arg\max}\limits_{G_{ex} \subseteq G} \ I(f(\cdot),G_{ex},G)$$
>     where each explanation $G_{ex}^i$ has $n_i$ nodes, and the other nodes not in the explanation can be expressed as $\{G \backslash G_{ex}^i\}=\{v_{j}\}_{j=n_{i}+1}^n$. Shapley value, as a concept originating from cooperative game theory, is used as an importance scoring function in this work. Therefore, the nodes or substructures or explanations (each explanation contains one or more substructures) can be regarded as a *player* in the game. The payoffs or utility function in the game is the Shapley value. The goal of our method is to find an explanation that can maximize the Shapley value.
>
> **Q2. The Shapley value is only computed approximately (as common).**
>
> Indeed, accurately obtaining the Shapley value has always been a very difficult problem [1]. In order to approach the exact solution of the Shapley value to a certain extent, we adopted the k-hop Shapley (see eqn. 3 in the main manuscript).
>
> **Q3. The search space cannot be covered completely.**
>
> Given more time, MCTS can explore the search space more thoroughly and return better results. We acknowledge that our approach is not guaranteed to find the global optimum when the given running time is short. However, as MCTS is an "anytime" algorithm, the longer it runs, the better the explanation result tends to be.
>
> **Q4. The implications of the Theorem 1 need to be explained better.**
>
> Thank you for your advice. We have restructured the article to emphasize the theoretical framework as the main contribution in this part of our study. In particular, we have broadened the scope of our theoretical exploration in juxtaposition with subgraphX and moved the toy examples from the appendix to the main body of the text. We hope this modification emphasizes the importance of this framework in specific application settings and enhances our exposition of theoretical optimality. It is worth noting that we are not comparing different bounds. For a given graph, its optimal interpretation should be the same under our mathematical framework. What we want to emphasize is that the expansion based algorithm can be closer to the optimal explanation than the pruning based algorithm. Please kindly find the detail in *Overall response: Q5*.
>
> **Q5. Missing literature: GraphSVX.**
>
> Thank you for sharing this related work; we will add a discussion of them to the related work section in final the version. In short, although GraphSVX and our work both use Shapley value as a value function, the technical aspect of our method is orthogonal to GraphSVX: SAME is a perturbation-based method while GraphSVX is a surrogate method.
>
> **Q6. No code.**
>
> We have released the code and sent an anonymized link to the AC in a separate comment.
>
> **Q7. Minor critique to Section 2.2 Line 110-121 and 130-134.**
>
> We apologize for confusing you with the content in Section 2.2. We have revised this section.
>
> We have modified the overall structure of Section 2.2: firstly, we introduce our mathematical framework; then, under this framework, we emphasize the difference between the pruning algorithm and the expansion-based algorithm, thus making our results clearer. Please kindly find the revised content in *Overall response: Q5*.
>
> **Q8. For what kind of GNNs does the proposed explainer work? Could attention be considered?**
>
> As the SAME is a *model-agnostic* technique, our approach does not need further adaptation or modification for application to different GNN models. Please kindly find the related experiments in *Overall response: Q3*.
>
> **Reference**
>
> [1] The shapley value in machine learning. 2022.

---

> > ### Comment · Reviewer_xcjE · 2023-08-14
> > **Score update**
> >
> > I thank the authors for their response and additional explanations. I have updated my score accordingly.

---

### Official Review · Reviewer_dDYx · 2023-07-05

**Soundness:** 1 poor
**Presentation:** 2 fair
**Contribution:** 2 fair
**Rating:** 5
**Confidence:** 4

**Summary:**

The paper introduces a novel method for explaining GNNs called SAME. SAME is theoretically grounded and has some good properties over exisiting methods. SAME uses an expansion-based Monte Carlo tree search algorithm to approximate the optimal Shapley-based explanation, which is proven to be better than pruning-based methods like SubgraphX. SAME has two MCTS stages, first on nodes and then on connected component combinations. SAME is evaluated on six datasets from different sources with diverse categories, including molecular graphs, sentiment graphs, and synthetic BA graphs. The experiments show that SAME outperforms previous state-of-the-art methods and has reasonable inference time.

**Strengths:**

1. A theoretical assessment of the expansion-based vs. pruning-based explanation methods is great.
2. The two-stage MCTS approach involving nodes and then components is novel to me and intuitively makes sense.
3. The proposed SAME framework has good empirical performance.

**Weaknesses:**

1. The level of detail for different parts of the paper can be adjusted

One suggestion I have is to shrink section 2.1 by moving some content to the Appendix. Although being clear about what Shapley value is and what the good properties of Shapley value have is important, most of the target readers should be comfortable reading the paper without mentioning all the details explicitly in the main text. For example, properties 1 - 4 should easily follow as long as the set of players P_i is clearly defined. Also, "k-hop Shapley" is another fancy term for saying only nodes in the computation graph should be considered. This idea was introduced in SubgraphX, and the idea of computation graph was brought up even earlier in GNNExplainer. All of these being said, the core idea of Section 2.1 can be quickly made clear by defining the set of players and the computation graph as k-hop neighbors. However, a whole page listing out all four properties and formulas (2) and (3) can make the paper seem redundant and suspicious, as these are not original contributions of the paper.

I would suggest spending more space discussing other important things. For example, at the end of Section 2, the authors mentioned that "It is intuitive to conclude ....". This kind of discussion can be made more detailed when there is more space.


2. Required properties of explanation methods can be made more clear

Several properties of GNN explanation methods are introduced in Table 1, and the major role of sections 2 and 3 is to show SAME is better than other methods in terms of these properties. However, I think some of these properties are not well-defined or not reasonably named, making them not the best ways to measure explanation methods.

a. Structure awareness. The authors claim that 4 methods satisfy this property and 3 do not. However, Section 2.1 discusses that the way SubgraphX and SAME satisfy the structure awareness is through the "k-hop Shapley" idea. I would not say "k-hop" is structure independent, but it only trivially uses the structure to construct the computation graph. The structure awareness refers to a different concept from the existing structure-aware idea proposed by GStarX, as the computation of HN naturally involves the structure. Thus, the property here can be misleading.

b. Multiple explanations. By multiple explanations, the authors seem to mean whether multiple disconnected pieces are allowed to appear together in the explanation output. However, the name here only makes sense when these pieces have a "or" relationship instead of an "and" relationship. For example, when the ground truth explanation for a prediction is "A and B". It is not proper to say there are multiple explanations, one being "A" and the other being "B", because neither of them is a complete explanation. Also, this multiple-explanation idea seems more like a limitation of the SubgraphX method instead of a property in general. SubgraphX assumed connectedness so there will only be one piece, which can be a good assumption instead of a limitation in some cases, but other methods do not make this assumption. In general, I do not think it is a good idea to take an assumption only one method makes and claiming not having it is a required property.

c. Node-wise/Substructure-wise/global importance. I don't feel this categorization makes full sense. If we just look at the definition, global importance implies substructure-wise importance, which further implies node-wise importance. This is because one explanation can be one connected component, which can be one node. How could one method have the global importance property but not the other two? Like GNNExplainer and PGExplainer in Table 1. Also, I believe that if any of these three importance properties is strictly satisfied, the computational complexity will be exponential. How could SAME satisfy them and be a polynomial algorithm?

I hope the authors can better organize these properties so that they 1. do not have conflicts with existing concepts 2. are properly named 3. are more clearly defined and categorized.

**Questions:**

My questions are very related to the weaknesses I mentioned above.

1. Why is it intuitive to conclude the expansion-based method is better than the pruning-based method? Can the authors explain more?

2. If I misunderstood the properties in Table 1 in my weaknesses comment, please point it out.

3. I have some doubts about the time complexity analysis. Can the authors be more detailed about why it is O((M1 + M2)n^2)? Also, other methods, like SubgraphX, do not necessarily have time complexity O(2^n), as SubgraphX considers k-hop neighbors, uses MC approximation, and has a budget for the subgraph size.

---

> ### Author Rebuttal · Authors · 2023-08-10
>
> Thank you for your careful review and comments. Below please find our point-to-point responses to your comments.
>
> **Q1. The level of detail for different parts of the paper can be adjusted.**
>
> Thank you for your excellent suggestions. We agree that we should adjust the detail for Section 2. We have shorter the description of Section 2.1 and provided more details about Section 2.2 following your advice. Please kindly find the revised version in *Overall response: Q5*.
>
> **Q2. Required properties of explanation methods can be made more clear.**
>
> Thank you for giving this thoughtful suggestion. The detailed definitions of the property in Table 1 are as follows.
>
> 1. Graph-level tasks: the GNN explanation method can handle the graph classification/regression tasks.
>
> 2. Node-level tasks: the GNN explanation method can handle the node classification/regression tasks.
>
> 3. Feature interactions: given a graph $G$, when measuring the importance of the explanation result $G_{ex}$, the GNN explanation method can consider the influence of {G/G_{ex}} on the importance of $G_{ex}$.
>
> 4. Structure awareness: given a graph $G$, when measuring the importance of explaining the result $G_{ex}$, the GNN explanation method is sensitive to the topology of the given input graph $G$.
>
> 5. Multiple explanations: the GNN explanation method can provide the explanation $G_{ex}$ that can be composed of one or more connected components.
>
> 6. Node-wise importance: given an input graph $G$ to be explained, for any node $v_i\in G$, its $I(f(\cdot),v_i,G)$ importance will be considered by the GNN explanation method.
>
> 7. Substructure-wise importance: given an input graph $G$ to be explained, for any substructure $G_{sub_i}\subseteq G$, its importance $I(f(\cdot),G_{sub_i},G)$ will be considered by the GNN explanation method.
>
> 8. Global importance: given an explanation $G_{ex}^i\subseteq G$ consisting of one or more substructures, its importance $I(f(\cdot),G_{ex}^i,G)$ will be considered by the GNN explanation method.
>
> 9. Priority-based integration: given an explanation $G_{ex}^j$ with any size, the node $v_i\in\{G\backslash G_{ex}^j\}$ will be added by the GNN explanation method on $G_{ex}^j$ to get a new explanation $G_{ex}^k$, if and only if for any $v_l \in \{G\backslash (G_{ex}^j\cup v_i)\}$, $I(f(\cdot),G_{ex}^j\cup v_i,G) > I(f(\cdot),G_{ex}^j\cup v_l,G)$ holds.
>
> 10. Redundancy consideration: given an explanation $G_{ex}\subseteq G$, if $I(f(\cdot), G_{ex}\backslash \{i\}, G)$ $>$ $I(f(\cdot), G_{ex}, G)$ holds, the new explanation $G_{ex}'=G_{ex}\backslash \{i\}$ will be chosen by the GNN explanation method.
>
> We agree that the **structure-awareness** mentioned by the reviewer is inconsistent between the ‘k-hop Shapley’ used in our work and HN-value mentioned by GstarX. The structure-awareness defined in our work is a macro definition. If the GNN explanation method can explicitly (e.g. GstarX) or implicitly (e.g. SubgraphX and ours) process structural information of the input graph, we call it has structure-awareness property.
>
> Your understanding of the **multiple explanations** is correct. We also think that subgraphX's method of finding connected substructures is a good assumption, which can make explanation results more in line with human intuition. Indeed, in the first phase of SAME, we also consider the connectivity of substructures when initializing important substructures set with expansion-based MCTS. However, it needs to be admitted that SubgraphX cannot find the explanation that the ground truth contains multiple substructures at the same time, which is one of their limitations. The lack of ability to find explanations where the ground truth contains multiple substructures will limit the usability of the method in many scenarios. Thus, we listed the *multiple explanations* in the property comparison table.
>
> For the **necessity of distinguishing between Node-wise / Substructure-wise / Global importance**, it is possible for one method to have the global importance property but not the other two. For example in GNNExplainer and PGExplainer, they obtain the explanation by learning a mask for the given graph. Therefore, the explanation results from them only consider the global importance but not substructure-wise importance or node-wise importance. It is also possible for one method to have the node-wise importance property but not the other two. For example, in GstarX, the importance of any substructure is simply the sum of the importance of the node without taking them as a whole to derive the importance. Therefore, it is necessary to distinguish among them.
>
> We admit that if one method is strictly satisfied any of the above three importance properties, it will lead to an NP-hard problem.  SAME, like other SOTA methods, uses an approximating algorithm to measure the Shapley value and explore the possible global optimal solution with the help of MCTS.
>
> Note that our approach is based on MCTS, but with a novel two-phase expansion-based MCTS version, which is shown to outperform the subgraphX, and outperform or be competitive with other SOTA methods (see Table 2, 3 in the main manuscript; Table S3, S4 in the supplementary material; *Overall response: Q4* for additional experiments for detailed results).
>
> **Q3. More details about time complexity of SAME. And, other methods, like SubgraphX.**
>
> We refine the computational complexity. Please kindly see the relevant answer to this question in *Overall response: Q1*. As SubgraphX is a pruning-based method, it requires more time in the simulation step, especially when the given graph is very large and the explanation is small.
>
> As Table 2 and Table 3 in the main manuscript, Table S3 and Table S4 in the supplementary material and additional experiments in *Overall response: Q4* show, our SAME significantly outperform SubgraphX in terms of different explainability metric with shorter inference time on different benchmarks.

---

> > ### Comment · Reviewer_dDYx · 2023-08-16
> > **Response to authors**
> >
> > 1. Thank the authors for adding the time complexity analysis. My current understanding is that the SAME algorithm is polynomial because of MCTS. I hope the authors make this clear in the final version. Otherwise, people may mistake that there is a polynomial algorithm for Shapley value computation.
> >
> > 2. For structure-awareness, I hope the authors can consider how to properly explain it to readers. As you mentioned, there is an "inconsistent between the ‘k-hop Shapley’ used in our work and HN-value mentioned by GstarX".
> >
> > 3. For the other properties, I don't feel my concerns are properly addressed at the moment. I will think about the response more carefully, and I suggest the authors read my original questions again.
> >
> > 3.1 Can the authors discuss more of my question about the "and" and "or" scenario?
> >
> > 3.1 By looking at the definition of property 8 and property 6 again. I still don't get why 8 doesn't imply 6. What if $G_{ex}^{i}$ is a single node? Maybe my misunderstanding is due to the statement "importance will be considered" being too vague. Hope the authors can explain more.

---

> > > ### Author Response · Authors · 2023-08-18
> > > **Thank you for your further suggestive comments**
> > >
> > > **1. Time complexity**
> > >
> > > Your understanding that the SAME algorithm is polynomial due to MCTS is correct. We will make this point explicitly clear in the final version of the paper.
> > >
> > > **2. Structure-awareness**
> > >
> > > We will add the following content in the main manuscript so that it will not cause misunderstanding.
> > >
> > > *“We call a GNN explanation method has the structure-awareness property if it can process the structural information of the input graph, whether through explicit mechanisms, as demonstrated by GstarX, or implicit modalities, as exemplified by SubgraphX and this study.”*
> > >
> > >
> > > **3. Concerns about "Multiple explanations" and "Node-wise / Substructure-wise / Global importance".**
> > >
> > > **3.1 Discussion about the "and" and "or" scenario in the "Multiple explanations" property.**
> > >
> > > The name of the "Multiple explanations" property can cause misunderstanding. Please allow us to clear this misunderstanding by modifying "Multiple explanations" into "Multi-piece explanation". The "Multi-piece explanation" means whether multiple disconnected pieces are allowed to appear together in the explanation output. We only consider "and" scenario, but not "or" scenario. For example, when "A" and "B" are two orthogonal connected components and the ground truth explanation for a prediction is "A and B", then the GNN explanation method has a probability to provide the correct "A and B" explanation.
> > >
> > > The following is why we think "Multi-piece explanation" is an important property of a GNN explanation method.
> > >
> > > Finding connected substructures in SubgraphX is a good assumption, which can make explanation results more in line with human intuition. After all, the explanation needs to be human-centred. `Indeed, in the first phase of SAME, we also search connected components when initializing important substructures set with expansion-based MCTS`.
> > >
> > > However, it needs to be admitted that SubgraphX cannot find the explanation when the ground truth explanation composes of multiple substructures. The lack of ability to find explanations where the ground truth contains multiple substructures will limit the usability of such a method in many scenarios. Thus, we treated the "Multi-piece explanation" as a promising property and listed it in the property comparison table.
> > >
> > >
> > > **3.2 Clarification about "Node-wise / Substructure-wise / Global importance".**
> > >
> > > We apologize for the misleading of the previous definitions about "Node-wise / Substructure-wise / Global importance". We have modified the name "Global importance" into "Composite-wise importance", and revised the detailed definitions of "Node-wise / Substructure-wise / Composite-wise importance". We hope that the following modifications will make the definition clearer, and we will put the revised version of all properties into the supplementary material.
> > >
> > > 1. **Node-wise importance**: given any node $v_i\in G$, its $I(f(\cdot),v_i,G)$ importance can be considered by the GNN explanation method.
> > >
> > > 2. **Substructure-wise importance**: given any substructure $G_{sub_i}\subseteq G$, it can be directly calculated as a whole through the GNN explanation method to obtain its importance $I(f(\cdot),G_{sub_i},G)$. Note that any substructure is a connected component which has more than one node.
> > >
> > > 3. **Composite-wise importance**: given any composite $G_{com}^i\subseteq G$ consisting of more than two nodes or substructures, it can be directly calculated as a whole through the GNN explanation method to obtain its importance $I(f(\cdot),G_{com}^i,G)$.
> > >
> > > Under the above definition, a). it is possible for one GNN explanation method to only have the composite-wise importance property but not have the other two properties. For example, in GNNExplainer and PGExplainer, both of them obtain the explanation by learning a mask with a `predefined size` for the given graph. The explanation results from them only consider the composite-wise importance but not substructure-wise importance or node-wise importance.  b). It is also possible for one method to only have the node-wise importance property but not have the other two properties. For example, in GStarX, the importance of any substructure is the `sum of the importance of all nodes in the substructure` without taking them as a whole to derive the importance. Therefore, it is necessary to distinguish among different importance.
> > >
> > > In this work, the proposed SAME can consider all node-wise importance (in the first phase). For substructure-wise and composite-wise importance, we admit that covering the search space corresponding to composite-wise importance or substructure-wise importance is an NP-hard problem. This paper provides a method to obtain a high-quality explanation by searching important substructures (in the first phase) and important composites (in the second phase) in polynomial time using expansion-based MCTS. Note that, with enough time, our method has the opportunity to cover the entire structure-wise and composite-wise search space.

---

> > > > ### Comment · Reviewer_dDYx · 2023-08-18
> > > > **Response to authors and score update**
> > > >
> > > > Thank the authors for their clarification response. As I mentioned to the AC, I think SAME has its strength but some of these clarity issues cause misunderstanding. Especially if you want to overload an existing terminology or mathematically define properties.
> > > >
> > > > I think changing "importance will be considered" to "can be directly calculated as a whole" is an important clarity improvement that helped me to better understand the properties.
> > > >
> > > > With these changes, I will raise my score to 5. Thank the authors for their response again.

---

### Official Review · Reviewer_8xQs · 2023-07-06

**Soundness:** 3 good
**Presentation:** 3 good
**Contribution:** 2 fair
**Rating:** 6
**Confidence:** 4

**Summary:**

The paper introduces SAME, a novel method for post-hoc explanation of Graph Neural Networks (GNNs). SAME leverages an expansion-based Monte Carlo tree search to explore structure-aware connected substructures and provides explanations that are as explainable as the theoretically optimal Shapley value. Experimental results show that SAME outperforms previous state-of-the-art methods, improving fidelity performance by 7.01% to 42.3% across various benchmarks. The SAME method contributes to the field of GNN explanation by offering theoretical foundations and improved performance.

**Strengths:**

- Theoretical foundation: The paper introduces the SAME method, which provides a theoretical foundation for post-hoc explanation techniques for Graph Neural Networks (GNNs). This adds credibility and rigor to the proposed approach.

- Structure-aware explanations: SAME addresses the challenge of structure-aware feature interactions in GNN explanations. By leveraging an expansion-based Monte Carlo tree search, SAME explores multi-grained connected substructures, resulting in more informative explanations.

- Improved performance: Experimental results demonstrate that SAME outperforms previous state-of-the-art methods in fidelity performance across various benchmarks. The improvements range from 7.01% to 42.3%, indicating the effectiveness of the proposed method.

**Weaknesses:**

- Limited discussion of limitations: The paper lacks a comprehensive discussion of the limitations or potential drawbacks of the SAME method. Addressing the potential shortcomings would strengthen the paper and guide future research directions.

- Limited comparison with alternative methods: While SAME is shown to outperform previous state-of-the-art methods, the paper does not provide a thorough comparison with alternative approaches. Including such a comparison would enhance the understanding of how SAME stands against other techniques.

- Generalizability to different GNN architectures: The paper does not explicitly discuss the generalizability of the SAME method to different GNN architectures. Considering and discussing the potential challenges or adaptations needed for different architectures would enhance the applicability of the proposed method.

- Theoretical complexity and computational efficiency: Although SAME claims to provide explanations within polynomial time, the paper does not provide a detailed analysis of the theoretical complexity or computational efficiency. Further investigation into the computational aspects would be valuable for understanding the scalability of the method.

**Questions:**

- Theoretical Analysis: Could you provide more details about the theoretical foundation of the SAME method? Specifically, how does the expansion-based Monte Carlo tree search contribute to the structure-aware explanations, and how does SAME ensure the explanations are as explainable as the theoretically optimal Shapley value?

- Comparative Analysis: In the paper, SAME is shown to outperform previous state-of-the-art methods in terms of fidelity performance. However, could you provide a more thorough comparison with alternative explanation techniques? How does SAME compare to other state-of-the-art methods in terms of interpretability, computational efficiency, and generalizability?

- Limitations and Future Directions: While the results are promising, it would be helpful to have a more comprehensive discussion on the limitations of the SAME method. What are some potential drawbacks or challenges associated with SAME? Additionally, what future research directions do you envision to address these limitations and further improve the method?

- Applicability to Different GNN Architectures: The paper focuses on explaining Graph Neural Networks (GNNs). Could you provide insights into the generalizability of SAME to different GNN architectures? Are there any specific considerations or adaptations required for applying SAME to other GNN models, such as graph attention networks or graph convolutional networks?

- Theoretical Complexity and Scalability: The paper claims that SAME provides explanations within polynomial time. Could you provide more details on the computational complexity of the SAME method? How does the method scale with increasing graph size or complexity? Are there any limitations or considerations regarding the scalability of SAME to larger graphs?

- Insights on Parameters and Hyperparameters: The paper mentions the use of Monte Carlo tree search and distinct single substructures in the SAME method. Could you elaborate on how the parameters and hyperparameters of these components affect the quality of the explanations? Are there any guidelines or insights you can provide to help researchers fine-tune these parameters effectively?

**Limitations:**

- Limited Discussion on Practical Applicability: The paper primarily focuses on the fidelity performance of the SAME method without discussing its practical applicability or real-world constraints. It would be beneficial to address any potential limitations or challenges in applying SAME to real-world scenarios, such as scalability, interpretability, or sensitivity to noise or perturbations in the input graph.

- Lack of User Evaluation: The paper does not include user studies or evaluations to assess the usefulness or understandability of the explanations generated by SAME. Incorporating user feedback or conducting user studies would provide insights into the effectiveness of SAME from a human-centered perspective.

- Absence of Ablation Studies: The paper does not include ablation studies to analyze the contribution of individual components or design choices in the SAME method. Conducting such studies would help isolate and evaluate the impact of different aspects of the method on the overall performance.

- Limited Dataset Coverage: The paper evaluates SAME on a specific set of benchmarks, both real-world and synthetic. However, it would be valuable to assess its performance on a more diverse range of datasets, including those with different graph sizes, properties, and domains.

---

> ### Author Rebuttal · Authors · 2023-08-10
>
> Thank you for your careful review and comments. Below please find our point-to-point responses to your comments.
>
> **Q1. The theoretical foundation of SAME. How does the MCTS contribute to the Structure-awareness, and how does SAME ensure the theoretically Optimal?**
>
> 1. To our best knowledge, no existing work has considered or formalised the theoretically optimal value of MCTS. The core of our theoretical work is to establish a framework for MCTS-based GNN explanation techniques. Under this framework, we define an error bound that can be widely applied to different MCTS-based methods. We show that our method has lower error bounds than another MCTS-based GNN explanation technique (SubgraphX). In order to explain our theoretical work more clearly, we adjusted the content of Section 2. Please see *Overall response: Q5* for detailed a explanation. Therefore, although in section 2.2, we did not consider enough theoretical foundation in the design part of the algorithm, we realized the "nature closer to the optimal solution" reflected in the comparison between expansion-based algorithms and pruning-based algorithms.
>
> 2. Because we use the **k-hop Shapley value** in the expansion-based MCTS, the interaction between interpretation and neighbor nodes will be considered in the calculation process, so as to achieve structural awareness.
>
> 3. In our theoretical work, we define an error bound that can be applied to different MCTS-based methods. We show that SAME has lower error bounds than another MCTS-based GNN explanation technique (SubgraphX). Here we adjusted the content of Section 2. Please see *Overall response: Q5* for detailed a explanation.
>
> **Q2. More thorough comparison in terms of interpretability, computational efficiency, and generalizability?**
>
> 1. For **interpretability** and **computational efficiency** of our approach. We compared our method with competitive techniques (see Table 2, 3 in the main manuscript, Table S3, S4 in the Supplementary Material).
>
>    We add new experiments on Twitter and BACE with additional results on GraphSVX and OrphicX (*Overall response: Q4*).
>
> 2. For **generalizability** of our approach, we further validate SAME on Graph-SST2 & Graph-SST5 using GIN and GAT (*Overall response: Q3*). In short, SAME does not need further adaptation on different GNN architectures since it is a post-hoc model-agnostic technique.
>
> **Q3. A detailed analysis of the computational efficiency.**
>
> We refine the computational complexity. Please kindly see the relevant answer in *Overall response: Q1*.
>
> **Q4. Limited discussion of limitations of SAME.**
>
> Please kindly find the related answer in *Overall response: Q2*.
>
> **Q5. Generalizability of SAME? Specific adaptations for GNN like GAT or GCN?**
>
> As the SAME is a post-hoc model-agnostic technique, our approach does not need further adaptation or modification for application to different GNN models. For the **generalizability** of our approach, please kindly refer to *Overall response: Q3*.
>
> **Q6. How the parameters and hyperparameters affect the quality of the explanations? Are there any guidelines or insights?**
>
> The hyperparameters include $\gamma$ and $K$. We randomly select 30 graphs from Graph-SST5 datasets using GCN, and report the fidelity w.r.t. $\gamma$ and $K$ in the following table.
>
> | $\gamma$    \    $K$ |    1    |    3    |    5    |    7    |    9    |
> | :------------------: | :-----: | :-----: | :-----: | :-----: | :-----: |
> |          1           | 0.25114 | 0.33188 | 0.35484 | 0.39379 | 0.40260 |
> |          2           | 0.28510 | 0.31207 | 0.34179 | 0.36521 | 0.37856 |
> |          3           | 0.29380 | 0.31802 | 0.32966 | 0.36365 | 0.37412 |
> |          4           | 0.31396 | 0.33553 | 0.34911 | 0.36104 | 0.35396 |
> |          5           | 0.31396 | 0.33869 | 0.35042 | 0.35909 | 0.35917 |
>
> Although $\gamma$=1 could lead to an optimal fidelity, the explanation may be composed of disconnected nodes, making the it not human-understandable. Therefore, we fine-tuned $\gamma$ and $K$ according to the visualization results. The detailed settings are provided in the Table below.
>
> |   Datasets   | BA-2Motifs | BBBP | Graph-SST2 | Graph-SST5 | BA-Shapes | MUTAG | Twitter | BACE |
> | :----------: | :--------: | :--: | :--------: | :--------: | :-------: | :---: | :-----: | :--: |
> | **$\gamma$** |     3      |  5   |     3      |     3      |     5     |   2   |    3    |  2   |
> |   **$K$**    |     7      |  7   |     7      |     7      |     7     |   7   |    7    |  7   |
>
> **Q7. Limitations in applying SAME to real-world scenarios, such as scalability, interpretability, or sensitivity to noise or perturbations in the input graph.**
>
> Thank you for your careful review and suggestions. We answer your questions point-to-point below.
>
> 1. We supplement the discussion about **scalability** of our method, please find the related content in *Overall response: Q2*.
>
> 2. In terms of the limited discussion about **interpretability**, we beg to differ in that the explanatory benchmark includes synthetic, sentiment and molecular datasets (in Table 2). We also considered different two kinds of fidelity as our metric (in Table S3, S4).
>
>    Note that for some data the explanation with the maximum fidelity cannot match the ground truth. Therefore, when applying to a real-world dataset, it is necessary to use different metrics with actual needs.
>
> 3. Considering the **sensitivity to noise or perturbations in the input graph** is not generally an issue for the post-hoc graph explanation technique. Because if a graph explanation technique is robust to noise, it may overestimate or underestimate the well-trained GNN that needs to be explained.
>
> Due to the character limit of the response, for other limitations, please find the corresponding responses in the following comments.

---

> > ### Author Response · Authors · 2023-08-16
> > **Additional response to Limitations 2-4 (Q8-Q10)**
> >
> > **Q8. Incorporating user feedback or conducting user studies would provide insights into the effectiveness of SAME.**
> >
> > Thank you for your excellent suggestion. Indeed, to the best of our knowledge, existing GNN explanation methods do not incorporate human feedback in the search for explanations. This is an excellent future direction in the field.
> >
> > **Q9. No ablation studies to analyze the contribution of individual components or design choices in SAME.**
> >
> > SubgraphX can be treated as our ablation version as they only do substructure searching (phase 1) but not our explanation set searching (phase 2). As Tables 2, 3 Table S3 and S4 show, our method outperforms SubgraphX in terms of different metrics with shorter inference time.
> >
> > **Q10. Assess performance on a more diverse dataset with different graph sizes, properties, and domains.**
> >
> > We added new experiments on Twitter [5] and BACE [6] (*Overall response: Q4*). The diversity of graph size in different benchmarks is summarized as follows.
> >
> > |   Datasets   | BA-2Motifs | BBBP | Graph-SST2 | Graph-SST5 | BA-Shapes | MUTAG | Twitter | BACE |
> > | :----------: | :--------: | :--: | :--------: | :--------: | :-------: | :---: | :-----: | :--: |
> > | **Min Size** |     25     |  2   |     1      |     2      |    700    |  10   |    3    |  10  |
> > | **Max Size** |     25     | 132  |     56     |     56     |    700    |  28   |   73    |  97  |
> >
> > **Reference**
> >
> > [5] Explainability in graph neural networks: A taxonomic survey. 2022.
> >
> > [6] MoleculeNet: a benchmark for molecular machine learning.2018.

---

### Official Review · Reviewer_qWxr · 2023-07-10

**Soundness:** 3 good
**Presentation:** 2 fair
**Contribution:** 2 fair
**Rating:** 6
**Confidence:** 2

**Summary:**

The paper proposes a novel method for explaining GNNs called SAME. SAME addresses the challenges of structure-aware feature interactions in GNN explanation by using an expansion-based Monte Carlo tree search. The authors evaluate SAME on a variety of benchmarks and show that it outperforms previous state-of-the-art methods. The paper makes contributions to the field of GNN explanation.

**Strengths:**

- SAME addresses the challenges of structure-aware feature interactions in GNN explanation by using an expansion-based Monte Carlo tree search, which is novel
- The authors evaluate SAME on a variety of benchmarks, and the experiments show that SAME outperforms previous state-of-the-art methods on these benchmarks.
- The theoretical aspects of SAME are well-developed and provide a good foundation for the proposed method.


**Weaknesses:**

- The authors could provide more information on the limitations of SAME, particularly in terms of the types of graphs and datasets for which it may not perform as well.
- The authors could provide more insight into the computational complexity of SAME and how it scales to larger graphs and datasets.
- The technical content is not sufficient, for example, it does not fully use the 9-page limit.

**Questions:**

How scalable is SAME? Shapley-based explainability methods are generally more expensive than gradient-based methods.

**Limitations:**

Yes.

---

> ### Author Rebuttal · Authors · 2023-08-09
>
> Thank you for your time and thorough comments! Below please find our point-to-point responses to your comments.
>
> **Q1. More insight into the computational complexity of SAME and how it scales to larger graphs and datasets.**
>
> For providing insight into the computational complexity and scalability of SAME, we refine the computational complexity. Please kindly see the relevant answer to this question in *Overall response: Q1*.
>
> **Q2. More information on the limitations of SAME, particularly in terms of the types of graphs and datasets for which it may not perform as well.**
>
> Please kindly find the related answer to this question in *Overall response: Q2*.
>
> **Q3. How scalable is SAME? Shapley-based explainability methods are generally more expensive than gradient-based methods.**
>
> As we discussed in our answer to *Overall response: Q2*, our method may take a long time to get an explanation on large graphs, which may be more expensive in time than gradient-based methods.
>
> However, would like to point out that the proposed SAME has obtained the SOTA explanation results (see Table 2 in the main manuscript: fidelity = 0.214±0.000) on the current explanation benchmark with the largest graph size (see Table below: BA-Shapes with size = 700) in a relatively short period of time (see Table 3 in the main manuscript: inference time = 14.08s). This indicates that our method has the possibility to work well on larger graphs.
>
> Table. The min / max graph size in different datasets.
> | Datasets | BA-2Motifs | BBBP | Graph-SST2 | Graph-SST5 | BA-Shapes | MUTAG | Twitter | BACE |
>    | :--------: | :----------: | :----: | :----------: | :----------: | :---------: | :-----: | :-------: | :----: |
>    | **Min**  | 25         | 2    | 1          | 2          | 700       | 10    | 3       | 10   |
>    | **Max**  | 25         | 132  | 56         | 56         | 700       | 28    | 73      | 97   |

---

> > ### Comment · Reviewer_qWxr · 2023-08-21
> >
> > I have read the authors' responses and I appreciated the efforts made by the authors. I plan to keep my current rating of 6 for the submission.

---

### Author Rebuttal · Authors · 2023-08-09

Dear Area Chairs and Reviewers,

We appreciate the valuable feedback and suggestions from the reviewers. Overall, the reviewers deem our paper well written, our method "novel" (qWxr,dDYx) and "effective" (8xQs), our theoretical analysis "well-developed" (qWxr), our evaluation results "standard" (xcjE). They also asserted that the problem studied in our paper "makes contributions to the field of GNN explanation" (qWxr, 8xQs).

In the following content, we make a general response to the questions that several reviewers are concerned about.

**Q1. More insight and details about the computation complexity of SAME.**

To better analyze the limitation (discuss in Q2) of our proposed method, we refine the computational complexity of our approach as follows.

Given a graph $G=(V,E)$, we take a single iteration as an example which includes selection, expansion, simulation and backpropagation [1].

For the **important substructure initialization phase**, the goal of our SAME is to figure out a group of important substructures in *Important Substructure Set* whose sizes are smaller than $\gamma$. In the *selection step*, our method chooses an unvisited node or chooses the node with the highest reward if all nodes have been visited ($\mathcal{O}(|V|)$). In the *expansion step*, our method selects the node within 1-hop neighbors with the largest value, which requires $\mathcal{O}(1)$. In the *simulation step*, in the worst case SAME will append all the nodes by visiting all the edges in the graph which requires $\mathcal{O}(|V|+|E|)$. For the *backpropagation step*, time complexity is bounded by $\mathcal{O}(\gamma)$, as the maximum size of a substructure (depth of the search tree) is $\gamma$. Since we perform $M_1$ iterations, according to the law of multiplication, the time complexity of the first phase $\mathcal{O}(M_1\gamma|V|^2+M_1\gamma|V|\times|E|)$.

For the **explanation exploration phase**, similar to phase 1, in the *selection step*, the time complexity requires $\mathcal{O}(K)$ since the size of *Important Substructure Set* equals $K$. The *expansion step* requires $\mathcal{O}(1)$. In the *simulation step*, our method will append other substructures to the current state until the explanation size exceeds the sparsity limit, which is bounded by $\mathcal{O}(2^{K})$. We set a maximum simulation time $t_s$ as a time budget that requires $\mathcal{O}(t_s)$ since $\mathcal{O}(2^{K})$ can be very large. The *backpropagation step* is bounded by $\mathcal{O}(\frac{|V|\times(1-sparsity)}{\gamma})$, where $\gamma$ is the maximum size of each substructure. With $M_2$ iterations, the time complexity in this phase is $\mathcal{O}(M_2 K t_s \frac{|V|\times(1-sparsity)}{\gamma})$.

Overall, the time complexity of SAME is $\mathcal{O}(M_1\gamma|V|^2+M_1\gamma|V|\times|E| + M_2 K t_s \frac{|V|\times(1-sparsity)}{\gamma})$. As the $t_s$, $\gamma$, $K$, $M_1$, $M_2$ and $sparsity$ are predefined, therefore SAME is a polynomial-time method under these constraints.

**Q2. More information about limitations of SAME.**

The limitations of our approach can be summarized as follows.

1. Scalability on large graphs. The time complexity of our method is $\mathcal{O}(M_1\gamma|V|^2+M_1\gamma|V|\times|E| + M_2 K t_s \frac{|V|\times(1-sparsity)}{\gamma})$.
    For searching large graph ($|V| $ is large) explanation,
    a. $|V|^2$ will become large.

    b. the $K$ needs to be larger accordingly to find the related explanation.

    c. the $t_s$ needs to be increased so that MCTS can converge.

    d. when the input graph is dense (|E| is large), $|V|\times|E|$ will also become large.

2. Approximation of Shapley value. For time efficiency, the Shapley value can only be computed approximately. Under this approximation, the fairness axioms no longer hold. This is also identified as an unsolved issue for the Shapley value in machine learning by [2].

**Q3. Generalizability of SAME on different GNN architectures**

Our method does not need further adaptations on different GNN architectures.

For the generalizability analysis, we tested SAME on GAT and GIN. The results are provided in Table 1 in the uploaded pdf. The results of the other two alternative explanation techniques (GraphSVX [3] and OrphicX [4]) are also provided.


**Q4. Further comparison of our SAME and alternative explanation techniques on other benchmarks**

To make our evaluation more convincing, we add new experiments on the text sentiment network Twitter [5] and molecular network BACE [6]. The results are presented in Table 2 in the uploaded pdf.

In the following individual response, we address all the raised questions and add some new experiment results to further strengthen our contributions.

**Reference**

[1] A survey of monte carlo tree search methods. 2012.

[2] The shapley value in machine learning. 2022.

[3] Graphsvx: Shapley value explanations for graph neural networks. 2021.

[4] Orphicx: A causality-inspired latent variable model for interpreting graph neural networks. 2022.

[5] Explainability in graph neural networks: A taxonomic survey. 2022.

[6] MoleculeNet: a benchmark for molecular machine learning.2018.

---

### Author Response · Authors · 2023-08-10
**Additional Overall response to Q5. Modification to Section 2.1 and Section 2.2**

**Q5. Modification to Section 2.1 and Section 2.2**

We will move the four properties and formulas (2) and (3) in Section 2.1 to the appendix to leave enough space for Section 2.2. Defining the optimal solution for the MCTS problem, irrespective of the reward mechanism employed, has always posed significant challenges.

In short, the primary motivation behind constructing this framework was to provide a more nuanced description of the nature of the search space, $X$. This detailed portrayal serves a dual purpose: on one hand, it must offer insights into the inherent characteristics of the 'optimal solution,' on the other hand, it must highlight the differential outcomes of various search methodologies. Inspired by [7], we have incorporated ample recursive logic and descriptive language to ensure that readers gain a lucid understanding of the specific objectives of each mathematical element introduced within this framework. The introduction of  $l$  is to specifically quantify the jumpiness of the contribution of each node in the search space (specifically represented by the size of the Shapley value), to realize the characterization of the GNN explanation method's ability to provide the multiple connected substructures simultaneously.

Back to the search space itself, we use a $K$-ary tree to specifically express the process of its segmentation. Each leaf of this $K$-ary tree represents the result of cutting the original search space $X$ (that is, the result of executing the search process), and the cutting result is called a cell$X_{h,i}$. The four assumptions are some sufficient and necessary conditions for the construction of the framework, and these conditions also satisfy the properties of the search space we described. Lemma 1 and Theorem 1 are necessary proofs of mathematical properties under this framework. By building this mathematical framework, we can achieve the two goals mentioned above. within this framework, we describe the properties of the optimal solution for the corresponding GNN interpretation method. We are thus able to illustrate how the expansion-based algorithm is closer to the optimal solution than the pruning-based algorithm.

**Reference**

[7] On the value of side-observations. 2011.

---

### Author Response · Authors · 2023-08-10
**Additional response to Limitations 2-4 for Reviewer 8xQs**

**Q8. Incorporating user feedback or conducting user studies would provide insights into the effectiveness of SAME.**

Thank you for your excellent suggestion. Indeed, to the best of our knowledge, existing GNN explanation methods do not incorporate human feedback in the search for explanations. This is an excellent future direction in the field.

**Q9. No ablation studies to analyze the contribution of individual components or design choices in SAME.**

SubgraphX can be treated as our ablation version as they only do substructure searching (phase 1) but not our explanation set searching (phase 2). As Tables 2, 3 Table S3 and S4 show, our method outperforms SubgraphX in terms of different metrics with shorter inference time.

**Q10. Assess performance on a more diverse dataset with different graph sizes, properties, and domains.**

We added new experiments on Twitter [5] and BACE [6] (*Overall response: Q4*). The diversity of graph size in different benchmarks is summarized as follows.

|   Datasets   | BA-2Motifs | BBBP | Graph-SST2 | Graph-SST5 | BA-Shapes | MUTAG | Twitter | BACE |
| :----------: | :--------: | :--: | :--------: | :--------: | :-------: | :---: | :-----: | :--: |
| **Min Size** |     25     |  2   |     1      |     2      |    700    |  10   |    3    |  10  |
| **Max Size** |     25     | 132  |     56     |     56     |    700    |  28   |   73    |  97  |

**Reference**

[5] Explainability in graph neural networks: A taxonomic survey. 2022.

[6] MoleculeNet: a benchmark for molecular machine learning.2018.

---

### Comment · Area_Chair_mVJM · 2023-08-17
**AC discussions**

There are mixed reviews for this paper.

Reviewers: could you read author rebuttals and let us know if your concerns have been addressed?

Reviewer dDYx: could you please respond?

AC

---

> ### Comment · Reviewer_dDYx · 2023-08-17
> **Already responded. Concerns still not addressed**
>
> I responded yesterday. I think both the AC and the authors should be able to see my responses? Please let me know if my responses are not visible.
>
> So far I haven't heard back from the authors. I read the reviews by other reviewers as well, and I do agree that this paper has its strength. However, my concerns regarding the property definitions and connection to some related works are not addressed. I hope the authors can further explain.

---

### Decision · Program_Chairs · 2023-09-21

**Decision:**

Accept (poster)

**Comment:**

This paper received consistently positive reviews, and major issues have been resolved during rebuttals.